# River-bed armouring as a granular segregation phenomenon

Behrooz Ferdowsi[1,2,3], Carlos P. Ortiz[2,4], Morgane Houssais[5] & Douglas J. Jerolmack[2]

River bed-load transport is a kind of dense granular flow, and such flows are known to segregate grains. While gravel-river beds typically have an "armoured" layer of coarse grains on the surface, which acts to protect finer particles underneath from erosion, the contribution of granular physics to river-bed armouring has not yet been investigated. Here we examine these connections in a laboratory river with bimodal sediment size, by tracking the motion of particles from the surface to deep inside the bed, and find that armour develops by two distinct mechanisms. Bed-load transport in the near-surface layer drives rapid, shear rate-dependent advective segregation. Creeping grains beneath the bed-load layer give rise to slow but persistent diffusion-dominated segregation. We verify these findings with a continuum phenomenological model and discrete element method simulations. Our experiments suggest that some river-bed armouring may be due to granular segregation from below—rather than fluid-driven sorting from above—while also providing new insights on the mechanics of segregation that are relevant to a wide range of granular flows.

[1] Department of Geosciences, Princeton University, Princeton, NJ 08544, USA. [2] Earth and Environmental Science, University of Pennsylvania, Philadelphia, PA 19104, USA. [3] National Center for Earth-surface Dynamics 2, University of Minnesota, Third Avenue SE, Minneapolis, MN 55414, USA. [4] Department of Physics and Astronomy, University of Pennsylvania, Pennsylvania, PA 19104, USA. [5] Benjamin Levich Institute, The City College of New York, New York, NY 10031, USA. Behrooz Ferdowsi and Carlos P. Ortiz contributed equally to this work. Correspondence and requests for materials should be addressed to D.J.J. (email: sediment@sas.upenn.edu)

River-bed grain size controls the exchange of solutes, nutrients and fine particulates across the sediment–fluid interface[1,2], and influences the flood magnitude required to initiate motion[3–5]. Grain size, however, also evolves over a series of floods as particles are sorted longitudinally and vertically during transport[3,6–8]. A ubiquitous pattern observed in gravel-bed rivers is armouring, in which the median grain size of the surface is significantly larger than that of the subsurface. Laboratory experiments designed to simulate gravel rivers—i.e., bed-load transport of heterogeneous grain sizes—reproduce the phenomenon, but are unclear on its origins. Three potential mechanisms have been proposed: kinetic sieving, in which smaller particles migrate downward through the void spaces between larger particles during motion[9]; "equal mobility", whereby the proportion of large and small surficial grains adjusts to achieve a spatially constant entrainment stress[6]; and sediment supply imbalance, in which the transport capacity of the flow locally exceeds the upstream supply and results in surface coarsening[7]. All of them assume that gravel in transport only mixes with the substrate over a small "active layer" that is one to several grain diameters deep.

Recently, sediment transport experiments have revealed that granular motion extends much deeper into the subsurface[10,11]. In particular, grains transition continuously from rapid bed-load motion at the surface to slow creeping motion far below the surface[11,12]. Both kinds of motion also occur in dry granular systems, where bed-load corresponds to a dense granular flow, and creep is characteristic of quasi-static deformation of disordered granular packs[11,12]. The former is known to produce robust vertical size segregation by kinetic sieving[13–15]. Phenomenological continuum models based on this premise[15–18] produce vertical segregation that is consistent with experimental observations[16,19] and discrete element method (DEM) simulations[20,21]. Segregation by creep is unexplored; while reports of slow coarsening do exist[22], its connection to creep has not been demonstrated.

The contribution of granular physics to river-bed armouring has only begun to be examined. Frey and Church[9] showed with laboratory experiments that bed load drives segregation by kinetic sieving that is qualitatively similar to dense granular flows. Here we investigate granular segregation and quantify its contribution to armouring using an idealized laboratory river experiment. Our setup is designed to: eliminate the disruptive influence of flume boundaries by using an annulus; image particle motion from the sediment–fluid interface to deep in the subsurface, away from the wall; isolate granular contributions by simplifying particles to bimodal spheres and eliminating fluid turbulence with a viscous fluid; and explore a range of transport conditions from near threshold to vigorous bed load. These experiments demonstrate how river-bed armour can develop due to bottom-up motion of subsurface grains, while revealing new insight on granular segregation mechanisms in a system where rapid and slow granular flows co-exist. Results are compared with predictions from a modified phenomenological segregation model, and with DEM simulations of a dry-granular bed under shear.

## Results

**Experiments.** Our experiments were conducted in a closed-top annular flume (Fig. 1); details of the apparatus have been described previously[11,12]. The channel walls are smooth to allow slip between grains and the boundary, in order to approximate an infinitely deep and wide channel. The flume is filled with a bidisperse granular bed of acrylic spherical grains with small and large diameters, $d_s = 1.5$ mm and $d_l = 3.0$ mm, respectively, and density $\rho_p = 1.19 \, g \, mL^{-1}$; the ratio of total small to large grain volume in the channel is $V_{small}/V_{large} = 2$. At this grain-size ratio $d_l/d_s = 2$, the ancillary phenomenon of spontaneous percolation of fines into the deep, quasi-static bed is likely to be negligible[23]. The grains are submerged in a fluid of viscosity $\eta = 72.2$ mPa s and density $\rho = 1.05 \, g \, mL^{-1}$. A fluid gap (clear fluid layer) is sheared from above by rotating the lid of the flume to apply a constant fluid–boundary shear stress, $\tau$ (Fig. 1c); it is reported here as dimensionless Shields number for the small grains, defined as $\tau_s^* = \frac{\tau}{(\rho_p - \rho)gd_s}$, where $g$ is gravity. The associated Shields stress for large grains as $\tau_l^* = \frac{d_s}{d_l}\tau_s^*$. For reference, Shields numbers for each experiment are compared to the critical Shields number, $\tau_c^*$, that is classically used to identify the onset of sediment transport. Our previously determined critical Shields number for a monodisperse bed of small grains, $\tau_{cs}^* \simeq 0.1$[11], is used here as the reference critical stress, recognizing that the actual value may differ in this bidisperse system[24]. We determined empirically for the present experiments that the range of Shields numbers $\tau_{cs}^* \leq \tau_s^* \leq 5\tau_{cs}^*$ corresponds to bed-load transport: a thin surface layer of moving grains in frequent contact with, and supported by, an underlying granular bed that is creeping[11]. We report data from experiments conducted at five Shields numbers,

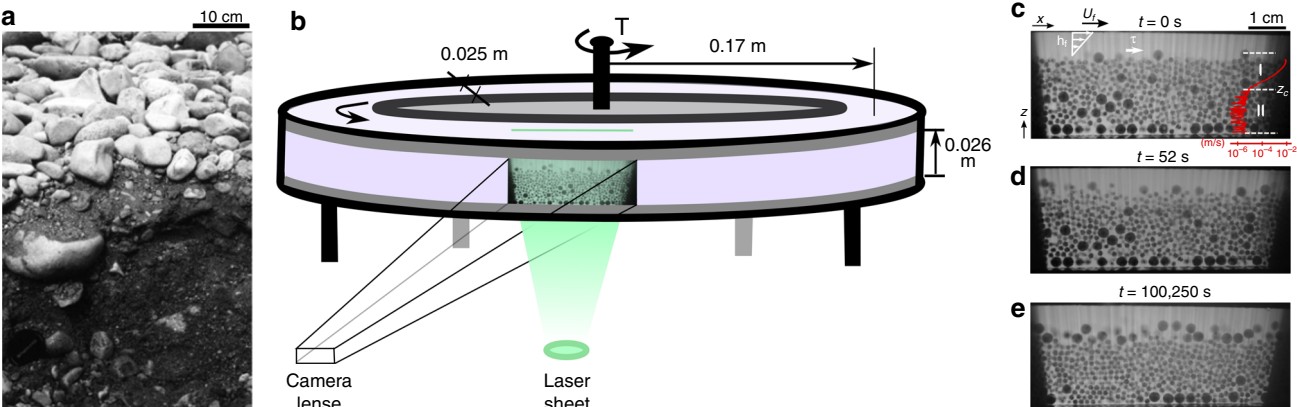

**Fig. 1** Phenomenology and setup. **a** Bed sediment of the River Wharfe, U.K., that shows a pronounced surface armour. Photo courtesy D. Powell[58]. **b** Sketch of the experiment, showing position of the camera and laser plane used for imaging inside the granular bed. **c–e** Snapshots during armour development for $\tau_s^* = 3.8\tau_{cs}^*$. Also shown is the fluid boundary stress, which is computed as $\tau = \eta U_f/h_f$[11]. $U_f$ and $h_f$ are the top-plate speed and flow depth, respectively. The red curve shows the long-term-averaged streamwise particle velocity $u_x(z)$, where I and II correspond to the bed load and creep zones, respectively. The directions $x$ and $z$ are indicated

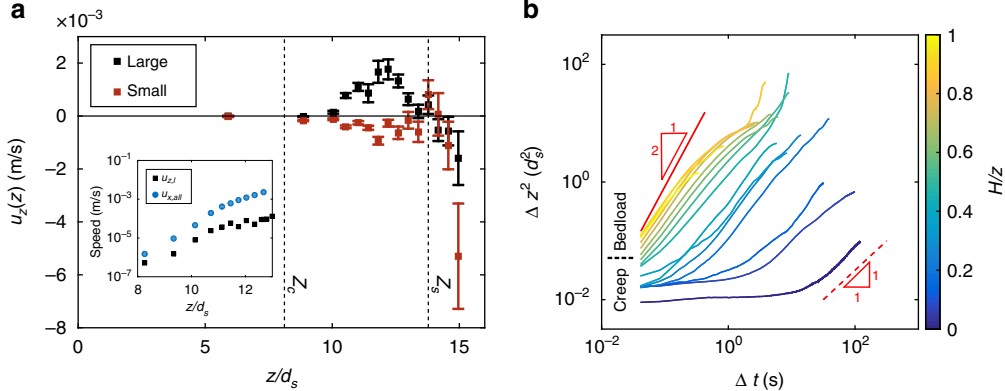

**Fig. 2** Experimental particle and segregation dynamics. **a** Vertical velocity profile for small and large grains for the interval $\Delta t = [0:20]$ min at the beginning of shearing at Shields number $\tau_s^* = 4.1\tau_{cs}^*$. The elevations of the bed surface ($z_s$) and transition to creep ($z_c$) are indicated. Inset shows the horizontal streamwise velocity $u_x$ profile for all grains and vertical velocity for large grains $u_{z,l}$ in logarithmic scale for the bedload zone. The streamwise particle velocity measurements have error-bar (one standard deviation/mean) value ~0.3%, whereas vertical particle velocity measurements have error-bar (one standard deviation/mean) value ~1%. **b** Vertical mean-square displacement (MSD) for large grains as a function of time $\Delta t$. MSD shown for various depths of the granular bed, defined with a colorbar. The top and bottom red-dashed lines indicate the limiting behaviors of advection and diffusion, respectively. Boundary between bed load and creep is indicated. Note that near-surface grains are advective at short times; creeping grains show no change in MSD at short times indicating caged dynamics, but transition to diffusive behavior at longer times

$\tau_s^* = [2.7, 3.8, 4.1, 4.4, 4.7]\tau_{cs}^*$. All flows were laminar (Reynolds number ≤ 4) and grain collisions were viscously damped (Stokes number < 1) (see Methods, section experimental setup and protocol).

The bed at the start of each experiment was composed of sedimented particles forming an approximately flat granular bed (see Methods). At the beginning of an experiment ($t = 0$ s) fluid shear was initiated at the specified Shields stress, and applied for a duration of 24 h or longer. We image a cross-section of particles from the bed surface ($z_s$) to the bottom of the channel through time using refractive-index matched scanning[25] (Fig. 1b; Supplementary Fig. 2). Vertical profiles of streamwise particle velocity ($u_x(z)$) for experiments at all Shields stresses were determined from averaging pixel strips in the streamwise direction over all time, using image cross-correlation (Methods, section velocity profiles). Velocity profiles confirm the existence of two distinct regions of particle motion (Fig. 1c). Zone I corresponds to bed load, where velocity decays rapidly with depth below the surface; below this is zone II associated with creep, characterized by a much slower decay[11]. All runs show a qualitatively similar evolution of the bed through time: a coarse surface "armour" layer develops as large grains are delivered from below; first more rapidly by bed load, and then more slowly by creep (Fig. 1, Supplementary Movies 1 and 2). This is explored in more detail below.

In order to probe the size- and depth-dependent behavior of grain motion, and its contribution to vertical segregation, we construct trajectories of all imaged grains using the particle tracking method[11] (see Methods) for a representative experiment at Shields stress $\tau_s^* = 4.1\tau_{cs}^*$. Profiles of average vertical velocity for large ($u_{z,l}$) and small ($u_{z,s}$) grains, computed from these trajectories, show a striking pattern: they are anti-correlated in the bed-load regime, with net upward (positive) velocity for large grains and net downward (negative) velocity for small grains. Although there are deviations in the near-surface (within $1d_s$ of $z_s$) due to intermittent saltation, below this region the vertical velocity of large grains decreases with depth and reaches approximately zero at the transition from bed load to creep ($z_c$) (Fig. 2a). The decay rate of $u_{z,l}$ is roughly exponential, and coincides with the decay of the bulk streamwise velocity $u_x$ (Fig. 2a-inset). This suggests that the observed vertical advection

of larger grains is linked to horizontal granular shear in the bed-load zone.

Grains in the creep zone have a small but detectable vertical velocity. To determine the dominant modes of particle motion in bed load and creep, we inspect the scaling of the mean-square displacements (MSD) vs. time. For the same experiment at $\tau_s^* = 4.1\tau_{cs}^*$ we compute the vertical MSD as a function of depth for the large grains as $\mathrm{MSD}(\Delta t) \equiv \Delta z^2(t) = \langle |z(t + \Delta t) - z(t)|^2 \rangle$ over a duration of 20 min; the brackets indicate ensemble averaging over grains and the reference time $t$, and $z$ is the particle's vertical position (Fig. 2; see Methods). A distinction can be made between grains above and below the depth associated with the transition from bed load to creep. Grains in the bed-load zone exhibit MSD growth at short times that approaches ballistic motion, and is consistent with the advection described earlier (Fig. 2a). The strength of the advection behavior diminishes at larger timescales where it perhaps transitions to super-diffusive behavior. In contrast, grains in the creep zone appear to exhibit caged dynamics in which MSD grows slowly or not at all at short timescales. Motion transitions toward diffusive and sometimes super-diffusive dynamics at longer times. The crossover timescale indicates the average lifetime of cages, and it increases with depth into the creeping zone. This behavior is similar to what has been observed in slow granular flows[26–28], and indicates that particle movement in creep is related to creation and destruction of the granular contact network[28,29].

To visualize the resulting development of surface armour, we examine the spatio-temporal concentration map of large grains; $\phi_l$ represents the streamwise-averaged areal fraction at a given depth and time (see Methods). The development of surface armour is seen as a high-concentration surficial layer that thickens through time (Fig. 3a; Supplementary Figs. 4–8). We quantify the thickness of the armour layer (Fig. 3a) as $z_{sa} - z_i$, where $z_{sa}$ and $z_i$ are the position of the top and bottom surfaces that define the armour layer, respectively (Methods, section determination of armour thickness) for all five Shields stress experiments. The data suggest the existence of two stages in the creation of armour, anticipated by the granular dynamics described above (Fig. 4a). First is rapid segregation (duration of $10^2$–$10^3$ s), as large grains are delivered up from the shallow subsurface. The rate of segregation shows a strong dependence on the driving Shields number, consistent with shear rate-dependent

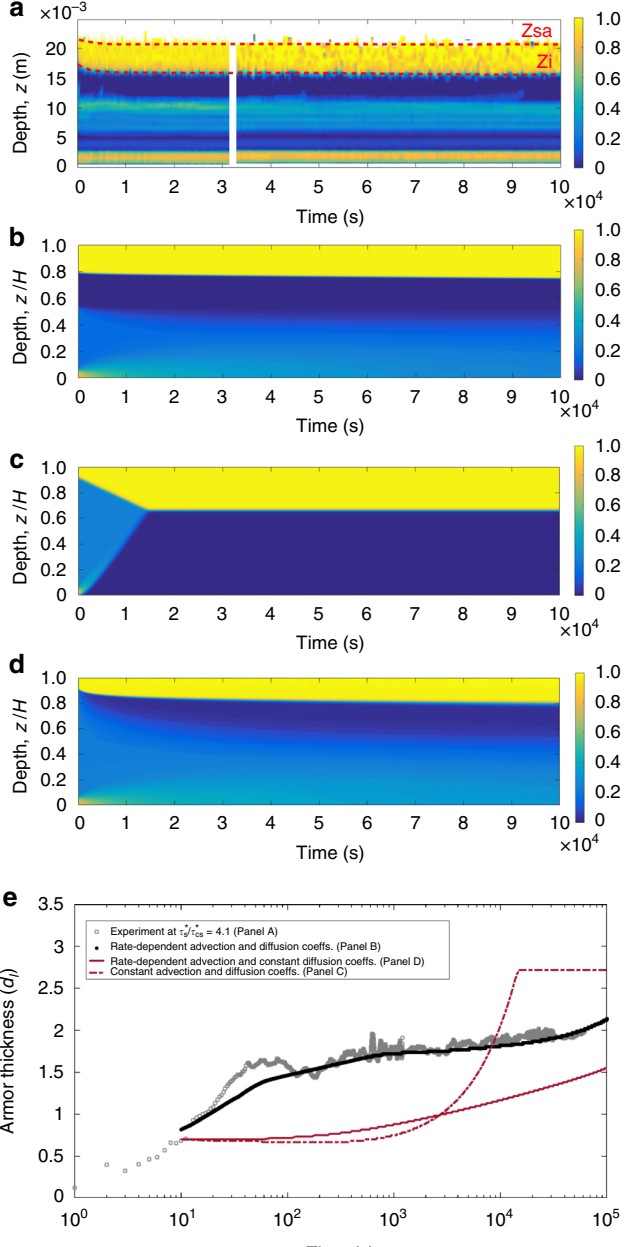

**Fig. 3** Evolution of coarse grain concentration and armour through time. **a** 1D (x-averaged) concentration map of large grains over time for shear stress $\tau_s^* = 4.1\tau_{cs}^*$. The red dashed lines show the positions of the armour surface ($z_{sa}$) and the bottom of the armored layer ($z_i$), which are used to calculate the thickness of the armored layer in Fig. 4a. **b** Same as **a** for the advection-diffusion model, using velocity profiles and initial conditions that correspond to the shear stress $\tau_s^* = 4.1\tau_{cs}^*$ in **a** and Fig. 2. **c** Concentration map of large grains from the advection-diffusion model using constant advection and diffusion coefficients. **d** Concentration map of large grains from the advection-diffusion model using the experimentally-determined advection profile, but a constant diffusion coefficient with depth. **e** Temporal evolution of the thickness of the armored layer at shear stress $\tau_s^* = 4.1\tau_{cs}^*$ calculated from experimental observation corresponding to **a** and different implementations of the advection-diffusion model

number, suggesting that creep segregation does not depend strongly on the driving shear rate.

Armour development in our experiments results from a vertical flux (z direction) of coarse grains toward the bed surface. We quantify this segregation flux, J, as the time derivative of the number of large grains in the armored layer:

$$J = A \frac{d}{dt} \int_{z_i}^{z_s} \phi_l dz \qquad (1)$$

where A is the cross-sectional area of the armour interface in the x−y plane. The variation of segregation flux density (J/A) with time (Fig. 4b) clearly shows the existence of two stages of armour formation. We introduce a dimensionless time $t/t_{adv}$, where the characteristic advection time $t_{adv} = \frac{h_{bl}}{\langle u_{z,l} \rangle} \sim \frac{h_{bl}}{a U_{sf}}$; $\langle u_{z,l} \rangle$, $U_{sf}$ and $h_{bl}$ are the average large grain vertical velocity, the average surficial grain velocity and the thickness of the bed-load layer, respectively, and $a = \langle u_{z,l} \rangle / U_{sf} \sim 10^{-3}$ is measured for the experiment at $\tau_s^* = 4.1\tau_{cs}^*$. We note however that this same value for a collapses all data later in Fig. 4c, indicating that this empirical coefficient is approximately constant for our experimental range. We also define a dimensionless segregation flux J/J(0) where J(0) is the initial value for J at the start of each experiment. Utilizing the dimensionless time and flux variables produces a reasonable collapse of the data (Fig. 4c). For all experiments J/J(0) decays to a value of 1/e at a characteristic dimensionless time of O(1).

**Advection-diffusion segregation model.** Sediment transport produces armouring that appears similar to reported granular segregation experiments[13,16], implying that the presence of a viscous fluid has little influence beyond determining the shear rate of surficial grains. In particular, some previous experiments in dry granular flows suggested that segregation rate depends on the granular shear rate[30,31], consistent with our findings for bed load (although another study found otherwise[19]). In addition, a recent study found that particle diffusion was shear rate-dependent for rapid granular flows but independent of shear rate for creep[32], similar to our experiments. Because the exact mechanism of segregation is still a subject of debate[18,21,33], there is no universally agreed upon continuum theory. Nonetheless, one-dimensional (1D) continuum models generally describe the vertical evolution of concentrations of binary mixtures through time with a phenomenological advection-diffusion equation[16,20]. Here we develop and apply a modified version of one such model, the Gray–Thornton model[15,16]. The model requires specification of: vertical advection and diffusion coefficients, usually assumed to be constant[15]; the vertical granular velocity profile; and the initial concentration profile. It then solves for the temporal segregation of large and small grains subject to mass conservation constraints. Two new ingredients must be included to account for the granular dynamics observed in our experiments: (i) for the bed-load regime, both advection and diffusion depend on shear rate; and (ii) for the creep regime there is no advection, and diffusion is independent of shear rate[32]. Our modified advection-diffusion segregation model, written in terms of the evolution of the large-grain concentration $\phi_l$, becomes:

$$\frac{\partial \phi_l}{\partial \hat{t}} + \frac{\partial}{\partial \hat{z}} (S_{rn}(\hat{z}) F(\phi_l)) = \frac{\partial}{\partial \hat{z}} \left( D_{rn}(\hat{z}) \frac{\partial \phi_l}{\partial \hat{z}} \right). \qquad (2)$$

Equation (2) is written in terms of dimensionless elevation $\hat{z} = z/H$ and time $\hat{t} = tU_{sf}/L$, where H and L are the height of the granular pack and the length of the centerline of the annular flume. The flux function $F(\phi_l)$ determines the dependence of the segregation flux ($S_r F(\phi_l)$) on $\phi_l$. Although there are ongoing debates on the mathematical form of the flux function[16,34], we

segregation of bed load. Once the bed-load zone is depleted of large grains, there follows a slow but persistent segregation that continues for the duration of the run (~24 h). We interpret the slow stage of segregation as creep driven. Interestingly, the rate of segregation in this stage is insensitive to the driving Shields

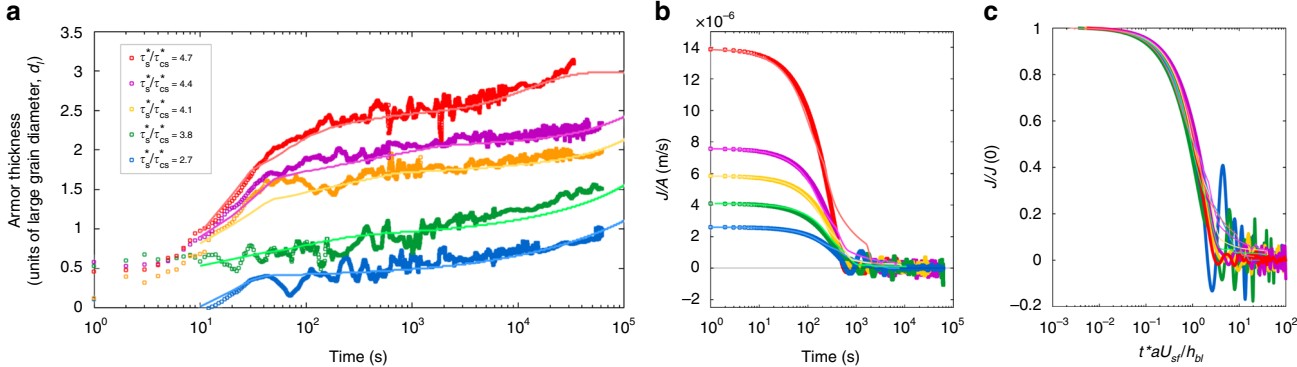

**Fig. 4** Armour thickness and segregation flux through time. **a** Temporal evolution of the thickness of the armored layer at different Shields number. Legend indicates Shields number associated with each curve, and applies to **b**, **c** also. The brighter continuous lines are predictions from the advection-diffusion model, Eq. (2). Note the first rapid stage of armouring, which is dependent on Shields number and is associated with bed-load transport, and the second slower stage that exhibits a nearly constant rate for all Shields numbers and is the result of creep. **b** The variation of segregation flux density ($J/A$) with time. **c** Normalized flux against dimensionless time; data are reasonably collapsed

implement the simplest choice: a quadratic function $F(\phi_l) = \phi_l(1 - \phi_l)$ that is symmetric about $\phi_l = 0.5$, which assumes that small and large grains behave identically but in opposite directions. The original Gray-Thornton model assumed a non-dimensional advective segregation velocity $S_r$ that is independent of shear rate. We introduce a depth-dependent parameter, $S_{rn}$, in order to redistribute the non-dimensional advective segregation velocity, $S_r$, according to the depth-dependent grain velocity, $u_x(\hat{z})$, normalized by the vertical average of grain velocities, $\langle u_x(\hat{z})\rangle$ (Eq. (3)).

$$S_{rn}(\hat{z}) = \begin{cases} S_r \frac{\beta \exp(\beta \hat{z})}{\exp(\beta) - 1} & : \hat{z} \geq \hat{z}_c \\ 0 & : \hat{z} < \hat{z}_c \end{cases} \quad (3)$$

The form of the normalized velocity is determined by a fit to the bed-load velocity profile such that:

$$\frac{u_x(\hat{z})}{\langle u_x(\hat{z})\rangle} = \frac{\dot{\gamma}_{xz}(\hat{z})}{\langle \dot{\gamma}_{xz}(\hat{z})\rangle} = \frac{\beta \exp(\beta \hat{z})}{\exp(\beta) - 1} \quad (4)$$

where $\dot{\gamma}_{xz}$ is the shear rate in $x - z$ plane for the bed-load layer, and $\beta$ is the exponential decay constant of the bed-load velocity profile. The values of $\beta$ vary with shear stress $\tau^*$ (see Supplementary Table 1), consistent with the local rheology of sediment transport[10–12]. For our analysis, we define the parameter $S_r = \frac{L}{H\langle u_x(\hat{z})\rangle} q$, where $q$ is the maximum bulk advective segregation velocity, i.e., that associated with the start of the experiment ($t = 0$; see Methods section implementation of continuum model; Supplementary Fig. 3). Similarly, we introduce a dimensionless and vertically-varying diffusivity $D_{rn}$ that has the same exponential decay as the velocity profile characterized by $\beta$ (Eq. (5)). The parameter $D_r = \frac{DL}{H^2\langle u_x(\hat{z})\rangle}$ is a non-dimensional diffusive-remixing constant, where $D$ is the dimensional diffusivity:

$$D_{rn}(\hat{z}) = \begin{cases} D_r \frac{\exp(\beta \hat{z})}{\exp(\beta) - 1} & : \hat{z} \geq \hat{z}_c \\ D_r \frac{\exp(\beta \hat{z}_c)}{\exp(\beta) - 1} & : \hat{z} < \hat{z}_c \end{cases} \quad (5)$$

To apply the new model Eq. (2) to our experiments requires specification of several parameters, determined from each experimental run (see Methods and Supplementary Figures). The input velocity profile $u_x(z)$ is determined by fitting two exponential functions to the time-averaged velocity profiles of the bed-load and creep zones, respectively (see Methods; Supplementary Fig. 6). The input value for $q$ is computed as the upward migration velocity of the center of mass for the large particles at

the start of each experiment (see Methods). Note that the advective segregation term in Eq. (3) decays with decreasing velocity (and depth) in the bed-load zone, and is set to zero in the creep zone ($z < z_c$). The diffusivity also decays with velocity (and depth) in the bed-load zone, and is constant for creep (Supplementary Fig. 6e). We take the dimensionless diffusion constant $D_r$ as a fitting parameter. In particular, the ratio $S_r/D_r$ is estimated by fitting the position of the armour interface through time for each experiment. We find that a constant ratio of $S_{rn}/D_{rn} \sim 318$ for the bed-load zone and $S_{rn}/D_{rn} = 0$ for the creep zone is sufficient to describe the development of armour for all Shields numbers. We use the profile of $\phi_l$ at the start of our experiments ($t = 0$) as the initial concentration profile for the continuum model (Supplementary Fig. 6f).

A visual comparison of armour development for the example condition $\tau_s^* = 4.1\tau_{cs}^*$ shows that the modified advection-diffusion segregation model (Eq. (2)) captures the experimental behavior well (Fig. 3b). We further run a model with constant advection-diffusion coefficient (same ratio $S_{rn}/D_{rn} \sim 318$) (Fig. 3c), and a model with depth-varying advection but a constant diffusion profile (Fig. 3d). Neither of the latter two models can reproduce the experimental observations. Our data are consistent with simulations by Fan et al.[32], who found that the diffusion coefficient in the dense-rapid granular flow regime is shear rate-dependent and becomes shear rate-independent in the creep regime. A more quantitative comparison of the thickness of the armored layer through time indicates that the model with rate-dependent advection-diffusion coefficients has superior predictive power (Fig. 3e), and matches the data well for the entire range of Shields stresses (Fig. 4a). Importantly, the model correctly captures the initial fast and subsequent slow stages of segregation. The large ratio $S_{rn}/D_{rn} \sim 318$ for $z > z_c$ confirms the idea that the rapid stage of armour development is driven by shear rate-dependent advection associated with bed load. The fact that the ratio $S_{rn}/D_{rn}$ remains constant for all experiments suggests that the model results are robust. The bulk kinetics can be related to particle-scale advection and diffusion by noting that $S_{rn}/D_{rn} = \frac{S_r}{D_r}\beta = Pe\frac{H}{d}\beta$, where the particle-scale ratio of advection to diffusion is given by the Peclet number $Pe = u_z d_l/D_z$. For the experiment with $\tau_s^* = 4.1\tau_{cs}^*$ we determined from measurements that $u_z = 1.51$ mm s$^{-1}$ and $D_z = 3.38$ mm$^2$ s$^{-1}$, which leads to $Pe = 1.3$ and $S_{rn}/D_{rn} = 140$; the latter is the same order of magnitude as the ratio used in the continuum simulations. The creeping zone is characterized by a constant value for $D_r$, and a lack of advection ($S_r = 0$), for $z < z_c$. This supports the notion that the slow stage of armouring results from diffusion by creeping

grains that is independent of local shear rate. Caution should be exercised in generalizing these results, however, as we have only performed experiments with a single bi-disperse and constant-density grain mixture in a single fluid.

**Discrete element modeling**. The analysis presented thus far shows how explicit accounting for the kinematics of granular motion in bed load and creep can produce a reasonable continuum description of armour development. In order to demonstrate that the observed armouring in experiments is entirely a consequence of granular physics, we now turn to DEM simulations in which the velocity profile and segregation dynamics arise spontaneously from grain-grain interactions. Simulations are performed with LIGGGHTS, an open-source granular modeling package based on LAMMPS (http://lammps.sandia.gov). Details of model implementation are available in Methods and Supplementary Information. In accord with the low Stokes number of our laboratory experiments, the restitution coefficient is chosen to be very small ($e_n = 0.01$ for Stokes number < 1, ref. [35]) such that collisions are highly damped (Supplementary Table 2). Otherwise, there is no treatment of the viscous fluid in DEM simulations.

The model domain is constructed to have a geometry, grain size and size-volume ratio that are the same as the experimental setup (Fig. 5a). The system is driven by a layer of large grains deposited at the surface and moving at constant velocity $u_{top}$ in the $x$ direction. Simulations are run for a duration that is equivalent to 1160 s and show behavior that is qualitatively comparable to the fluid-driven experiments of armouring, confirming the existence of two stages of segregation (Fig. 5). First is fast segregation within the rapid granular flow regime (first few grain diameters from the surface). Then, once grains are depleted from this "bed load" zone (Fig. 5c), armouring transitions to a slow stage driven by creep from deeper layers.

For a more direct comparison, we examine the growth of armour thickness through time (see Methods) for the previous simulation and an additional run with $u_{top} = 0.05 \, (\text{m s}^{-1})$ which corresponds to $\tau_s^* = 2.7\tau_{cs}^*$. For both runs the agreement of DEM simulations with the experiments is reasonably good (Fig. 5d). This agreement is especially encouraging given that: simulations neglect fluid flow entirely; the initial concentration of large grains in the experiments was difficult to control and not uniform; and there was no tuning or calibration done for the DEM runs, beyond adjusting the velocity of surface grains to match experiments.

Using DEM simulations, we can also explore the influence of changes in boundary roughness and submerged grain density—factors that may influence the dynamics we observe, but that are difficult to modify in the physical experiment. A rough channel bottom is simulated by attaching a fixed layer of hexagonally packed grains to the base. Results are not qualitatively different, in that we still observe a dense-granular flow regime that transitions with depth to a creeping regime; however, the rough boundary layer acts to slow the creeping velocity of grains near the bed (Supplementary Fig. 10). In addition, results of simulations with different grain density are qualitatively similar, indicating that bed-load and creep dynamics are robust (Supplementary Fig. 11).

## Discussion
Even though our flows were laminar, experiments and theory have shown that laminar bed load is similar to its turbulent counterpart in many respects[24,36–39]. Our results show armouring dynamics that are qualitatively similar to previous experiments[40,41] conducted under conditions more representative of

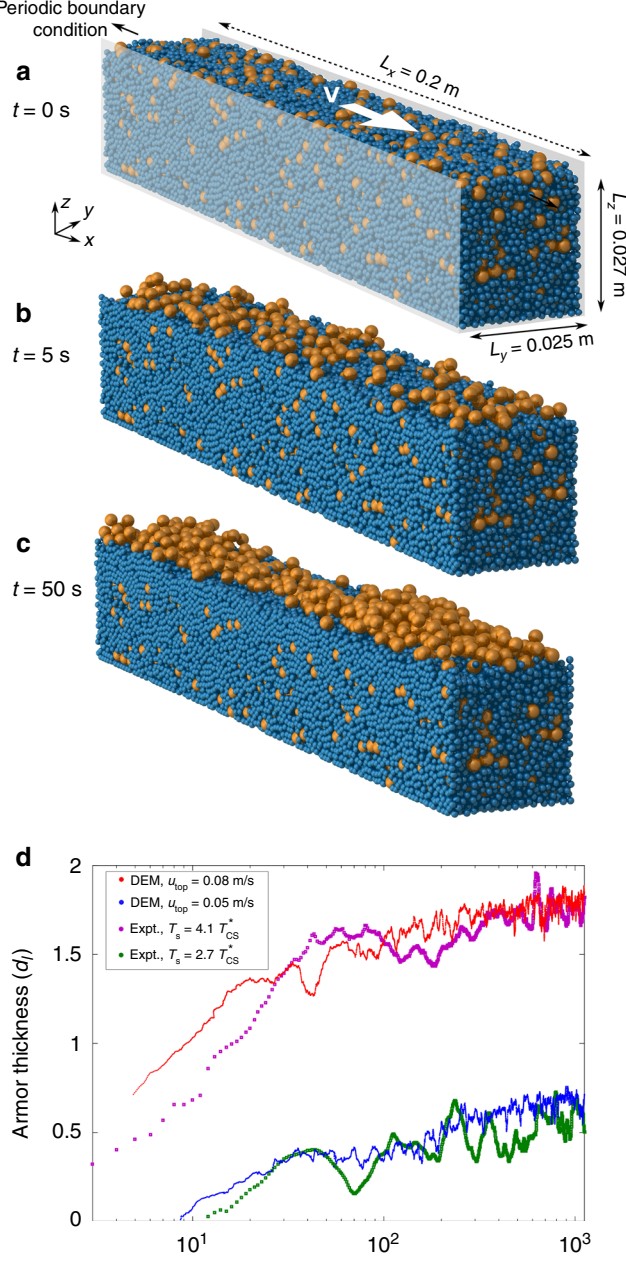

**Fig. 5** DEM simulation of a dry sheared granular bed with $u_{top} = 0.08 \, \text{m s}^{-1}$ equivalent to the fluid-driven sheared granular bed at $\tau_s^* = 4.1\tau_{cs}^*$. The top layer of large grains that drives particles underneath is shown in Supplementary Fig. 9. **a** Model domain and initial conditions. Granular pack shown after **b** 5 s and **c** 50 s of shearing. Note rapid segregation, and depletion of the near-surface zone of large grains, as a consequence of bed load. **d** Evolution of armour thickness for the DEM model and experiments at two equivalent driving stresses indicated in the legend

gravel rivers—i.e., poly-disperse and natural-shaped particles (average grain diameter $d \sim 1 \, \text{cm}$) in turbulent flows with driving stress $\tau^* \approx 2\tau_c^*$. Those studies[40,41] found a Shields-stress dependent armouring rate with a relatively rapid initial stage (a few hours) followed by slower stage. While data on particle motions were not reported, we can perform a scale analysis of the expected bed-load armouring timescale due to granular segregation, $t_{adv} \sim \frac{h_{bl}}{aU_{sf}}$, by assuming: $h_{bl} = (3-5)d$[42]; $U_{sf} \sim 1 \, \text{cm s}^{-1}$[43,44]; and our experimentally determined value $a \sim 10^{-3}$. This analysis

yields $t_{adv} \sim (1–2)$ h, within the observed range of experiments[40,41], and may be a reasonable bed-load armouring time-scale for natural gravel rivers. Translation to the field, however, may need to account for the presence of bed and bar forms that can influence armour formation[45]. Whether bars act primarily to increase the rate of vertical mixing, or introduce qualitatively new dynamics to segregation, is currently unknown.

Authors of previous experiments[7,40,41] attributed armour development to a lack of sediment supply to the channel, which they hypothesize resulted in winnowing of fines and concentration of coarse grains—in other words, sediment-supply imbalance. Our experiments, however, showed no significant size-selective transport at the surface and, more importantly, there were no supply limitations because the flume is annular. We can thus rule out sediment-supply imbalance for our experiments. Our results support the kinetic sieving model, on which the phenomenological Gray–Thornton equation is based. An important new finding, however, is that segregation does not occur only in the "active layer". If the bed-load zone corresponds to the active layer, then the associated sorting is important but occurs rapidly. Creep delivers grains from far below the bed-load zone to the surface, contributing to persistent armour development that was not previously recognized. The agreement of DEM simulations and experiments confirm the contention of Frey and Church frey[9] river that river-bed armouring may be considered to be a granular segregation phenomenon—at least in our idealized experiments. Results suggest minimal influence of the fluid beyond determining the surface grain velocity. We point out that sediment-supply imbalance may still be important for armouring in natural rivers; in particular, under sediment limitations such as downstream of dams where river beds experience net erosion that may preferentially remove finer grains[7]. We believe that the granular segregation dynamics revealed here, however, likely operate in all environments regardless of sediment supply and may therefore be more prevalent. Future experiments with more "natural" flow and particle conditions, that control for sediment supply while also examining precise granular motion from the surface to the bottom of the granular pack, would be helpful for assessing the relative importance of these different mechanisms. A potential field confirmation of the armouring mechanism proposed here would be the observation of a zone underneath the armour layer that is depleted of large grains (Fig. 3a, d). Size-selective surficial transport would not be expected to influence the concentration of coarse grains beneath the armour layer.

In our extended Gray-Thornton model, depth-dependent advection-diffusion parameters ($S_{rn}(z)$ and $D_{rn}(z)$) follow the variation of grain velocity in the granular bed with depth. It is noteworthy that parameter $\beta$ is not a calibration coefficient here, but it is the exponential decay constant of the fit to the variation of grain velocity with depth in each experimental run. Velocity in both bed load and creep regimes decays exponentially, but with different rate constants. Our model is based on the premise that advection and diffusion coefficients follow the same functional form and power of the velocity profile with depth in each experiment; therefore, the model inherently has a shear rate-dependency. In addition, since the derivative of an exponential function is the same exponential function, we can write $S_{rn}(z)$ and $D_{rn}(z)$ explicitly based on shear rate (rather than velocity). Doing so results in only a small change in constants $D_r$ and $S_r$, since the constant denominator in Eqs. (3) and (4), i.e., $(\exp(\beta) – 1)$, must be incorporated into the values of $D_r$ and $S_r$. The ratio $S_{rn}/D_{rn}$ remains the same for all stresses.

Our work sheds new light on the mechanics of granular segregation. Experiments clearly show that vertical advection of large grains is shear rate-dependent. Explicit accounting of this dependence, and also of shear rate-dependent diffusion, is needed

in order to explain observed segregation rates for the rapid granular flow regime (i.e., bed load). Moreover, data and models demonstrate that creep contributes to segregation, and that its mechanism is distinct from rapid granular flows. Large grains in the creep zone show no preference for upward- or downward-directed motion. Their long-time motion may be modeled as vertically isotropic and constant diffusion. Short-time dynamics show that creeping grains are caged, and indicate that their motion is likely induced by long-range transmission of forces through the granular contact network[46,47]. This may be why creep motion is independent of shear rate, at least for the range of Shields stresses examined here. It is intriguing that isotropic diffusion in creep can give rise to a net upward flux of large grains. Based on our results, we hypothesize that this flux arises because large grains that cross the boundary into bed load are then advected to the surface. If correct, this implies that a purely creeping granular pack (no bed load) should not produce armouring.

The experimental and modeling results presented here are a first step in assessing the contribution of fast and slow particle motion to vertical segregation. Our sediment mixture was bi-disperse in order to establish connections between granular shear segregation and river-bed armouring, but many systems of interest (including rivers) have a polydisperse grain size distribution that may exhibit different behavior. Such a distribution would challenge the application of continuum models, but is amenable to further experiments. River-bed armouring in our experiments and models was found to be driven by bottom-up granular segregation, rather than top-down surficial sorting driven by the fluid. Our findings show how information from the surface, in terms of fluid-driven shear, is transmitted deep into the subsurface through grain-grain interactions that are typically neglected in sediment transport models. Granular motion in the subsurface transmits information back to the surface through the delivery of coarse grains, linking surface dynamics to subsurface structure. By examining the river bed as a discrete medium, we were able to link the macroscopic pattern of armour development to the physics of sheared granular systems. Our results add to a growing body of evidence that sediment transport systems belong to a broader class of granular flows[9,11,12,48], and show how examining geophysical flows through the lens of granular physics can reveal novel insights for both fields. Extending our results to field settings, however, requires further work on the unexplored influences of fluid turbulence, bed and bar forms, and wider grain size distributions on the granular segregation mechanisms identified here.

## Methods

**Experimental setup and protocol.** Two dimensional (2D) images of our 3D experimental system were generated using index-matched PMMA particles ($d_s = 1.5$ mm, $d_l = 3.0$ mm, Engineering Laboratories) with a mixture of viscous oils (85% of PM550 and 15% of PM556 from Dow Corning). We dispersed dye (Exciton, pyrromethene 597) in the oil and excited it with a green laser sheet (517 nm, 50 mW) of thickness $\simeq d/10$. All experiments were conducted on a vibration-damping optical table and we used a damping coupler to connect the driving motor to the flume.

The bed preparation protocol was inspired by Golick and Daniels[30]. Grains were initially deposited in an inverse-segregated state, with large grains at the bottom, and then subject to a driving stress equivalent to $\tau_s^* = 20\tau_{cs}^*$ for ~1 min to fully suspend and mix the large and small populations. Fluid shear was halted and the suspension left for ~30 min to allow sedimentation, relaxation and compaction of the granular bed to reach completion (Supplementary Fig. 1). A second phase of slow relaxation and compaction starts once the actual shear experiment begins[49]. The duration of this phase in our previous monodisperse system study (which used a similar range of shear stresses) was found to be a few hours[12]. We cannot ignore the few first hours of the experiments that might be affected by this compaction, because segregation commences as soon as shear is applied. We cannot quantify the effects of compaction on the shear response of the granular bed and the rate of segregation, but we believe that the shear suspension preparation protocol decreases these effects. Moreover, we point out that the continuum model does not

include any influence of compaction, yet the fitted advection and diffusion values are comparable to those derived from grain-scale observations in the experiment. Supplementary Fig. 1 provides further information about the preparation protocol. The final random packed layer at the end of the preparation protocol had a thickness ~15.5$d_s$ for all experiments. After this first step, a constant rotation $\Omega$ is applied to drive the system for the entire duration of the experiments. This duration was not constant; each lasted long enough (24 h or longer) for all particles present in the recorded frames to exhibit some significant displacement during the run. We computed the fluid-flow depth $h_f = H_f - z_s$, where $H_f$ is the total depth of the flume and $z_s$ is the elevation of the surface as described below. We computed the fluid-flow velocity at the top plate in the channel center as $U_f = \Omega 2\pi R$, where $R$ = 17 cm is the radial distance to the channel center. The fluid boundary-shear stress is then calculated as $\tau = \eta U_f / h_f$. We assume in our entire analyses that the fluid flow is laminar and unidirectional in the azimuthal direction of the annular flume. We can justify the laminar assumption by showing that the Reynolds number associated with the fluid channel above the bed is small compared to turbulence flow limit. We estimate this Reynolds number as $Re = \frac{\rho U_{plate} h_f}{\eta}$, which is ≈4 for the largest $\Omega$ in our experiments reported here. The unidirectional assumption can be justified justified from the small ratio of radial viscous stress to the azimuthal viscous stress for our experimental conditions:

$$\frac{\text{Radial stress}}{\text{Azimuthal stress}} = cRe\frac{h_f}{R} = 0.4\% \qquad (6)$$

where $h_f \simeq 3d_s$, $R$ is the flume radius. In the equation above, $c \simeq 0.06$ is an estimated coefficient[36] that is only weakly dependent on the flow aspect ratio.

**Detection of the bed surface.** To detect the surface position, we first have calculated the concentration profile $C(z)$ for a given configuration of particles from a processed binary image. This binary image is valued at zero outside of particles and one inside of particles. The concentration is calculated next at each elevation $z$ pixel, as the pixel-wise average in the $x$ direction. As a result, this concentration profile can work as the one-dimensional analogue of packing fraction, which is the fraction of space occupied by the particles. The surface is defined from this concentration profile as the position $z_s$ where the concentration crosses fifty percent of its saturated value[11,50]. A fixed threshold of 0.35 is used here to define the surface position. The saturated value does not vary significantly in our set of experiments reported here. We define $z_s$ after averaging the concentration for a $\Delta t = 100$ s at the beginning of each experiment. This time duration is sufficiently long for the flux convergence time as observed in our earlier study[11]. Slow granular compaction[49,51] and slow dilation due to segregation[29] approximately counterbalance such that the surface position remains almost constant as the armouring experiments progress. The bed surface position is used for calculating the Shields stress at each experiment.

**Imaging technique and particle detection/tracking.** We used a Nikon DSLR 5100 digital camera to record the real-time positions of single particles by acquiring the fluorescence intensity from a laser dye (concentration ≈ 1 μM) dispersed in the fluid. The configuration is suitable for long data acquisition without significant photobleaching. The images were acquired continuously at 24 Hz for 10–20 min at the beginning of experiments and in order to sample fast dynamics near the surface. For this fast dynamics, the relevant timescale is the settling time of particles over their own diameter $d/v_{sed} = 0.68$ s. We acquire single images at a rate of one every 15 s for 24 h or longer and in order to sample slow dynamics in the system. Supplementary Fig. 2a shows a sample raw image at the start of an experiment. To detect the positions of the particles with subpixel accuracy, we find particle positions to pixel accuracy by peak-finding above a threshold. The details of the background correction process and further image processing are described in the Supplementary Materials of our previous publication[11] on monodisperse systems. A snapshot of detected particles with this method is superimposed on a gray-scale raw image and is shown in Supplementary Fig. 2b. The same detected particles are also shown in binary format in Supplementary Fig. 2c. A fixed diameter threshold of $d = 1.38$ mm is used for separating large and small particles in all experiments as shown in Supplementary Fig. 2d. Finally, a snapshot of identified large and small particles using this threshold is shown in Supplementary Fig. 2e. The local concentration of large grains is defined as $\phi_l(z) = \frac{\langle A_l \rangle_x}{\langle A_l + A_s \rangle_x}$, where $A_l$ and $A_s$ are large and small grains' projected areas in the imaged cross-section, respectively, and $\langle \cdot \rangle_x$ indicates pixel-wise streamwise integration.

**Velocity profiles.** For each experiment, a 6 min video capture with frame rate 24 fps at the start of the experiments is converted and processed into consecutive binary images following the procedure described in the imaging section above. The consecutive binary images, $I(t)$ and $I(t + \Delta t)$ are then used as the input of pixel-wise cross-correlation analysis along the $x$ direction at each pixel elevation $z$. The position of the central peak in the cross-correlation between $I(t)$ and $I(t + \Delta t)$ corresponds to the average streamwise distance traveled by grains at elevation $z$ during $\Delta t$ without regard to small and large particle species, i.e. for all particles. The result is averaged over the full duration of the video capture. This technique yields a time-averaged streamwise velocity profile $u_x(z)$ for all particles. Note, large particles are weighted more heavily than small particles. The results are in

agreement with velocity profiles determined from the particle-tracking method for all experiments. For the case of Fig. 2a, the velocity profile of large grains is computed using the particle tracking method described in the imaging technique section above.

**Determination of armour thickness.** The top surface of the armour layer, $z_{sa}$, is characterized as the position where the streamwise averaged concentration of large grains $\phi_l = 0.9$. The interface (bottom) of the armour layer with the rest of the granular bed, $z_i$, is calculated as the location where the gradient of $\phi_l$ reaches a minimum below the surface. The surface and interface positions time-series are smoothed using a running average of temporal window size 8.33 s for images obtained from video capture conversions and temporal window size 833 s for image captures. These are shown with dashed lines in Fig. 2c. The thickness of the armored layer is defined as $z_{sa} - z_i$.

**Implementation of continuum model.** The variables used to compute the advection and diffusion parameters for each experiment are reported in Supplementary Table 1. The maximum bulk segregation velocity, $q$, for each experiment is measured from relative displacement of the vertical $z$ component of the center of mass position of large particles $Z_{CM,l}$ relative to small particles $Z_{CM,s}$. The data for the relative $Z_{CM}$ displacement for all five stresses is presented in Supplementary Fig. 3. The initial concentration profile of large particles $\phi_l(0)$ is determined from the first 10 s of each experimental run; a simplified version of it is used as another input to the PDE model (Supplementary Figs. 4–8). The time-averaged streamwise velocity profiles for each experiment are reported in Supplementary Figs. 4–8 and are used to estimate the value of $\beta$. We use a numerical implementation of the method of lines solution to solve the PDE equations and use $N_{ts} = 10,000$ time steps for the full experimental time (~$10^5$ s). Comparisons of the concentration maps for the PDE model and experiments, for the four additional driving stresses, are presented in the Supplementary Materials (Supplementary Figs. 4–8).

**Implementation of DEM model.** The DEM model consists of a shear cell with sizes $0.027 \times 0.025$ m$^2$ in the $y \times z$ directions, and has a length 0.2 m in the $x$ direction where periodic boundary conditions are applied. The lateral sides in the $x$–$z$ plane and the lower boundary in the $x$–$y$ plane are smooth and frictional walls, with the same mechanical and frictional properties as the grains (Supplementary Table 2).

The top side in the $x$–$y$ plane is open. The cell is filled with $N = 38,812$ grains that are initially inserted randomly in the cell with a desired volume fraction of 0.45. It is then equilibrated under gravitational forces for 10 million time steps equivalent to $\Delta t = 20$ s. The initial concentration of large grains is uniform in the simulation domain. The grains are free to move in other directions (e.g., to dilate) in order to resemble a free-surface and shear-driven system. The grains are modeled as compressible spheres of diameter $d_{s,l}$ that interact when in contact via the Hertz–Mindlin model[52–54]:

$$F = \left(k_n\delta\vec{n}_{ij} - \alpha_n\vec{v}\vec{n}_{ij}\right) + \left(k_t\delta\vec{t}_{ij} - \alpha_t\vec{v}\vec{t}_{ij}\right) \qquad (7)$$

where the first term is total normal force, $\vec{F}_n$, and the second term is total tangential force, $\vec{F}_t$. In Eq. (7), $k_n$ and $k_t$ are normal and tangential stiffness respectively, $\delta$ is the overlap between grains, $\alpha_n$ is the normal damping, $v$ is the relative grain velocity, $\vec{n}_{ij}$ is the normal vector at grain contact, $\vec{t}_{ij}$ is the tangential vector at grain contact, and $\alpha_t$ is the tangential damping. The full model implementation is available on the LAMMPS/LIGGGHTS webpage and several references[55–57]. In accord with the low Stokes number of our laboratory experiments, the restitution coefficient is chosen to very small ($e_n = 0.01$) such that collisions are highly damped (Supplementary Table 2). Otherwise, there is no treatment of the viscous fluid in DEM simulations. The DEM model system is frictional, meaning that the coefficient of friction, $\mu$, is the upper limit of the tangential force through the Coulomb criterion $F_t = \mu F_n$. The tangential force between two grains grows according to non-linear Hertz–Mindlin contact law until $F_t/F_n = \mu$ and is then held at $F_t = \mu F_n$ until the grains lose contact. The values of density, grain diameter, Poisson's ratio and acceleration due to gravity are chosen to match the experimental conditions. The values for coefficient of restitution and friction coefficient are chosen to mimic the effects from interactions with the fluid. The Young's modulus of the particles used here is chosen to be low (Supplementary Table 2), MPa rather than GPa, in order to increase the calculation time step and decrease computational cost; however, since the system is not under significant confining pressure, a softer grain–grain interaction will not have considerable effect on the results, and the simulation remains in the hard-sphere limit. The particles are sufficiently hard that we find no additional rescaling of time is necessary. The damping coefficients $\alpha_n$ and $\alpha_t$ are determined within the implementation of LIGGGHTS from the chosen value for the restitution coefficient, $e_n$.

**Influence of boundary roughness and grain density.** We also performed DEM simulations with inclusion of a roughness layer at the base of the granular bed (Supplementary Fig. 10), to determine whether this has a significant influence on creep dynamics. The roughness layer is made of a layer of hexagonal close packed large grains fixed at their positions. Comparison of the horizontal velocity profile in

simulations with and without the roughness layer shows that the general transition from bed load to creep is qualitatively similar. The magnitude of the horizontal velocity in the creep regime may decrease by an order of magnitude in the vicinity of the rough boundary, which indicates a higher dissipation rate at the boundary. In addition, we explored the influence of grain density (relative to fluid). Supplementary Fig. 11 shows the velocity profiles ($u_x(z)$), in DEM simulations of a bidisperse sheared bed, for different grain densities ($\rho_{PMMA}$, $\rho_{glassbead}$, $\rho_{PMMA}$ − $\rho_{fluid}$, for PMMA, submerged PMMA and glass, respectively) sheared with surface layer velocity $u_{top} = 0.05$ m s$^{-1}$. These results show that there is no qualitative change in the dynamics with change in the grain density (relative to fluid). We therefore believe that our observations are general.

**Data availability**. The experimental data that support the findings of this study are available in figshare project repository https://figshare.com/projects/River-bed_armouring_as_a_granular_segregation_phenomenon/24919 with identifiers dois:10.6084/m9.figshare.5421085; 10.6084/m9.figshare.5419528; 10.6084/m9.figshare.5419525; 10.6084/m9.figshare.5419522; 10.6084/m9.figshare.5419267; 10.6084/m9.figshare.5419339; 10.6084/m9.figshare.5419384; 10.6084/m9.figshare.5417863; 10.6084/m9.figshare.5417764; 10.6084/m9.figshare.5417761; 10.6084/m9.figshare.5417596. All DEM simulations were run with LIGGGHTS public version 3.2.1, released on 07-21-2015. Particle detection and image processing scripts can be obtained from previous project of Penn Sediment Dynamics (PennSeD) Laboratory accessible at doi:10.6084/m9.figshare.1269323.

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

## Acknowledgements

Research was supported by US Army Research Office–Division of Earth Materials and Processes grant 64455EV, US National Science Foundation (NSF) grant EAR-1224943, NSF INSPIRE/EAR-1344280, and NSF MRSEC/DMR-1120901. B.F. was a synthesis postdoctoral fellow of the National Center for Earth-surface Dynamics (NCED2 NSF EAR-1246761) when the research was carried out. B.F. also acknowledges support from the Department of Geosciences, Princeton University, in form of a Hess Fellowship.

## Author contributions

All authors contributed to experimental design. B.F. and D.J.J. wrote the manuscript with input and contribution from all authors. B.F. and M.H. performed the experiments. B.F. designed, run and analyzed numerical and computational (DEM) simulations. B.F. and C. P.O. analyzed data. All authors contributed to interpretation of experimental, numerical, and computational data. D.J.J. supervised the research and managed the project.

## Additional information

**Competing interests:** The authors declare no competing financial interests.

