## [Peer Review File · Nature Communications]

Reviewers' comments:

Reviewer #1 (Remarks to the Author):

Comments on Ferdowsi et al., River-bed armoring as a granular segregation phenomenon
Serious work on the mechanics of granular transport of grains of mixed size under fluid shear flows has only recently begun (apart from some pioneer work by R.A.Bagnold some 60 years ago). The primary ultimate objective of this work is to better understand 'bedload' transport of clastic sediments in rivers. The problem is complex and is best tackled in a series of reductionist experiments designed to reveal fundamental aspects of the phenomenon. Notable progress has been made by research groups led by Philippe Frey (IRSTEA, Grenoble), Eric Lajeunesse (IPGP, Paris) and, most recently, by Houssais and Jerolmack (present co-authors), with all groups borrowing relevant concepts from the more extensively studied topic of dry granular flows. This paper exposes for the first time (so far as I am aware) the significance of creep behaviour in the quasi-static bed under the relatively rapid shear layer at the bed surface, leading to upward displacement of relatively coarse grains to join the somewhat counterintuitive (but widely observed in nature) concentration of coarse grains on the bed surface. These observations are fundamental and important: as such, the paper deserves publication as expeditiously as possible.

The paper is well written and appears to me to need only minor revisions (see the annotated pdf attached). It also needs some reformatting – the references are strangely distributed amongst the figures. To answer your questions explicitly:

What are the major claims of the paper?

As stated in the abstract, the major original claims of the paper are that

"Creeping grains beneath the bed-load layer give rise to slow but persistent diffusion-dominated segregation. . . suggest[ing] that some river-bed armoring may be due to granular segregation from below rather than fluid-driven sorting from above."

Are they novel and will they be of interest to others in the community and the wider field?

The finding is novel, in context, and will certainly be of wide interest in the sediment transport community.

If the conclusions are not original, it would be helpful if you could provide relevant references.

There are antecedents in work on 'dry' granular mechanics (as implied by the authors' successful implementation of a developed DEM model to confirm their work). However, this does not detract from the novelty of the results for the sediment transport community, which has always thought of the substrate as absolutely static. The authors comprehensively document relevant prior work.

Is the work convincing, and if not, what further evidence would be required to strengthen the conclusions?

The work is convincing, and is made especially so by the successful modelling of the experimental results by two numerical approaches.

On a more subjective note, do you feel that the paper will influence thinking in the field?

This paper will definitely influence thinking in the field of clastic sediment transport by fluid shear flows. It also reports observations that will be of interest in the wider community of those interested in granular mechanics.

We would also be grateful if you could comment on the appropriateness and validity of any statistical analysis, as well the ability of a researcher to reproduce the work, given the level of detail provided.

I am not well qualified to be a close critic of the modelling support, but the results are all in order. More generally, I judge that the methods section gives sufficient information for the work to be satisfactorily reproduced, though possibly not exactly replicated, since granular experiments of this kind are notoriously influenced by myriad minor details, such as the quality of the glass flume walls and the surface finish of the experimental grains.

I greatly enjoyed reading this paper.

Michael Church

13 10 2016

Reviewer #2 (Remarks to the Author):

Referee report on « River-bed armoring as a granular segregation phenomenon » for publication in Nature Communications by B. Ferdowsi, C.P. Ortiz, M Houssais and D.J. Jerolmack

General comment

The manuscript present novel experimental data on grain size segregation in an annular shear cell. The grain size distribution is bidisperse and the flow is driven by a rotating lid. This apparatus gives rise to laminar bed-load transport conditions with Shields number always higher than the threshold of motion even for the largest particles. The results show that kinetic sieving occurs under bed load transport and lead to a rapid grain size segregation. This rather rapid process is dominated by advection. The experimental results also show that grain size segregation occurs in the creep layer but at a longer timescale than the one associated with the bed load layer. This slow process is dominated by diffusion. The nature of grain size segregation, advective or diffusive, is demonstrated by applying a modified Gray-Thornton model to the experimental conditions. Empirical advection velocity and diffusion coefficient are proposed by fitting the model results to the experimental data and lead to shear rate dependent coefficients. This is different from Gray and Thornton who used constant values for dry granular flows. The last part of the paper describes 3D Discrete Element Method numerical simulations corresponding to the experimental conditions. The numerical simulations have been ran for about 60 s and qualitatively reproduce the grain size segregation at short time scale.

The paper is well written and provide very valuable experimental data as well as modeling guidelines on grain size segregation under bed load condition. The comparison with the advection-diffusion model of Gray and Thornton is also very relevant and allows to demonstrate the similarity of grain size segregation under bed-load conditions with the dry granular case. It also highlights the differences such as the variable advection velocity and diffusion coefficient in space. The influence of the creep layer on the grain size segregation dynamic is quite new in the field of sediment transport and the authors data seems to show that this process could have an influence on river bed armoring at long timescales. I have more doubt about the significance of the DEM part which is rather qualitative. The authors could have done a lot more analysis on this part and it is quite frustrating for the reader at this point. I would recommend to either extend this part or remove it from the paper.

As a conclusion of my review, I believe the paper should be published after the following questions and comments are addressed by the authors.

Questions and comments

1) Concerning the extended Gray-Thornton model, the authors claims that the advection and diffusion coefficient in the bed-load layer should depend on the shear rate but in their proposition the authors relate these coefficient to the vertical position. The parameter beta of the exponential function for the advection-diffusion coefficients is calibrated for each condition and seems to be quite sensitive. The dependency on the shear rate is rather indirect however I believe the model would gain generality if these coefficients where directly parameterized based on the local shear rate. I think some clarifications are needed here.

2) In the same line, I would have liked to see the results of the Gray-Thornton model using constant advection-diffusion coefficient and also the sensitivity to the modeling of the creep regime using a constant diffusion coefficient. I wonder if the creep contribution is that important.

3) Similar to what the authors have done for the advection time scale in the bed load regime,

could the authors give an estimate of the time scale associated with the creep regime? This would help to estimate the influence of the creep regime to the segregation dynamic.

4) The conclusions on the scaling analysis should be taken carefully as only one set of bidisperse mixture has been studied so far. In order to get more insight into the scaling laws for segregation more combinations of particle diameters, density and fluid viscosity should be investigated. I think the authors should add a sentence in the discussion to warn the readers.

5) It is not obvious to me how sensitive the results are to the creep regime in terms of segregation? Could the authors clarify the relative importance of bed-load and creep regime for segregation at long time scales.

6) The choice of the restitution coefficient ($e_n=0.01$) should be justified based on the Stokes number. There exist some data in the literature that allows to do that (see Gondret et al. PoF 2002). This is only if the DEM part is kept in the future version of the manuscript.

7) In Fig 4D, it is hard to say that the DEM results agree with the experiments, at least for the creep influence on segregation as the model has not been run long enough. According to Fig 3, the creep influence occurs at time scale of about 10^4 s and the DEM run only last for 60s.

Minor comment

The legend of some figures are too small. I think the authors should try to improve the plots, there is a lot of informations in each figure, maybe they can split some of them.

Reviewer #3 (Remarks to the Author):

The velocity profiles in Figures S4 through S6 show that there is slip between the plastic lower wall of the annulus and the lowermost spherical particles. The previous article in Nature Communications (Supp. Fig. 7) by most of the same authors shows the same phenomenon. The rate of shear between the wall and particles is much greater than between the particles in the "creep" zone. In fact S4 and S5 show essentially zero mean shear rate in the creep zone (i.e. plug flow). Perhaps the slip is not surprising given that the buoyant weight of the particles are more than an order of magnitude less than sediment in water. The rate of slip between the wall and particles increases systematically with shear stress. It is likely that the slip is not continuous in time and space (stick-slip) and leads to jostling of particles above. This could be the reason for the scatter in particle velocities. I was not able to obtain the supplemental videos for this paper, but the movies from the previous Nature Communications article show that settlement and shear of grains in the interior over the first several hours. The authors did not address how this time dependent behavior was dealt with for averaging. Were the first several hours removed from the analysis?

The slip between the particles and bed is not addressed. I think it makes the conclusions of this paper (and the previous Nature Comm paper) dubious. The authors believe that the annulus setup to be superior to a flume with a downstream boundary. I have the opposite view. In fact, I would suggest installing ribs across the annulus wall to stop the slip. Another possibility is to use more dense spheres such as borosilicate. I recommend that the paper be rejected with encouragement of resubmission if the authors can fully address these issues.

Response to reviewers

We present in this document point-by-point response to reviewers' comments and suggestions with describing the changes made in the manuscript to implement those suggestions. The changes in the main paper are highlighted in reply to each point. Several significant additions have been made to the supplementary material, which are pointed to in the main text, in order to address questions raised by the reviewers. In response to questions about the influence of boundary roughness, we have run a new DEM simulation with a rough boundary in order to show that creep survives even when the bottom of the channel is not smooth. To address questions about the influence of grain density on creep, we have also run DEM simulations with 3 different grain densities. These model runs show that the qualitative dynamics and even the functional form of the segregation rates do not change. The new simulation results do not change any of our original conclusions — rather, they reinforce them by suggesting that the dynamics observed in our experiments are likely to be robust to changes in channel geometry/roughness and grain/fluid properties.

Reviewer Comments:

Reviewer #1:

Serious work on the mechanics of granular transport of grains of mixed size under fluid shear flows has only recently begun (apart from some pioneer work by R.A. Bagnold some 60 years ago). The primary ultimate objective of this work is to better understand “bedload” transport of clastic sediments in rivers. The problem is complex and is best tackled in a series of reductionist experiments designed to reveal fundamental aspects of the phenomenon. Notable progress has been made by research groups led by Philippe Frey (IRSTEA, Grenoble), Eric Lajeunesse (IPGP, Paris) and, most recently, by Houssais and Jerolmack (present co-authors), with all groups borrowing relevant concepts from the more extensively studied topic of dry granular flows. This paper exposes for the first time (so far as I am aware) the significance of creep behaviour in the quasi-static bed under the relatively rapid shear layer at the bed surface, leading to upward displacement of relatively coarse grains to join the somewhat counterintuitive (but widely observed in nature) concentration of coarse grains on the bed surface. These observations are fundamental and important: as such, the paper deserves publication as expeditiously as possible. The paper is well written and appears to me to need only minor revisions (see the annotated pdf attached). It also needs some reformatting – the references are strangely distributed amongst the figures. To answer your questions explicitly:

What are the major claims of the paper?

As stated in the abstract, the major original claims of the paper are that “Creeping grains beneath the bed-load layer give rise to slow but persistent diffusion-dominated segregation. . . suggest[ing] that some river-bed armoring may be due to granular segregation from below rather than fluid-driven sorting from above.”

Are they novel and will they be of interest to others in the community and the wider field?

The finding is novel, in context, and will certainly be of wide interest in the sediment transport community.

If the conclusions are not original, it would be helpful if you could provide relevant references.

There are antecedents in work on “dry” granular mechanics (as implied by the authors’ successful implementation of a developed DEM model to confirm their work). However, this does not detract from the novelty of the results for the sediment transport community, which has always thought of the substrate as absolutely static. The authors comprehensively document relevant prior work.

Is the work convincing, and if not, what further evidence would be required to strengthen the conclusions?

The work is convincing, and is made especially so by the successful modelling of the experimental results by two numerical approaches.

On a more subjective note, do you feel that the paper will influence thinking in the field?

This paper will definitely influence thinking in the field of clastic sediment transport by fluid shear flows. It also reports observations that will be of interest in the wider community of those interested in granular mechanics.

We would also be grateful if you could comment on the appropriateness and validity of any statistical analysis, as well the ability of a researcher to reproduce the work, given the level of detail provided.

I am not well qualified to be a close critic of the modelling support, but the results are all in order. More generally, I judge that the methods section gives sufficient information for the work to be satisfactorily reproduced, though possibly not exactly replicated, since granular experiments of this kind are notoriously influenced by myriad minor details, such as the quality of the glass flume walls and the surface finish of the experimental grains.

I greatly enjoyed reading this paper.

Michael Church
13 10 2016

Reply: We thank the reviewer for his positive remarks and encouraging and thorough review. We have implemented the minor corrections and suggestions proposed by the reviewer in his accompanying PDF file. A number of the corrections were grammatical or clarification remarks. *Three of comments* needed more details or additional references which are all implemented. The corrections and revisions are highlighted in the revised manuscript enclosed. For completion, we bring the response to those three comments also to these response letter:

1. Line 42: We agree with the reviewer that the three mechanisms proposed by previous researchers might be indeed different aspects of a single mechanical phenomenon. We therefor changed the "competing mechanisms" to "potential mechanisms" in the revised manuscript.
2. Line 80: We have added the proposed information by the reviewer that at these small ratio of large to small grain size, the ancillary phenomenon of spontaneous percolation of fines in the deep, quasi-static bed, is severely limited and is not a primary concern. We also added a citation for this observation.
3. Line 359: The local concentration of large grains is defined as, $\phi_l(z) = \frac{\langle A_l \rangle_x}{\langle A_l + A_s \rangle_x}$ where A_l and A_s are large and small grains projected areas in the imaged cross-section, respectively. We used projected areas, because imaging and particle detection in this study has been performed in 2D (x-z plane) and for a cross-section in the middle of the annular flume setup.

Reviewer #2:

Referee report on "River-bed armoring as a granular segregation phenomenon" for publication in Nature Communications by B. Ferdowsi, C.P. Ortiz, M Houssais and D.J. Jerolmack

General comment

The manuscript present novel experimental data on grain size segregation in an annular shear cell. The grain size distribution is bidisperse and the flow is driven by a rotating lid. This apparatus gives rise to laminar bed-load transport conditions with Shields number always higher than the threshold of motion even for the largest particles. The results show that kinetic sieving occurs under bed load transport and lead to a rapid grain size segregation. This rather rapid process is dominated by advection. The experimental results also show that grain size segregation occurs in the creep layer but at a longer timescale than the one associated with the bed load layer. This slow process is dominated by diffusion. The nature of grain size segregation, advective or diffusive, is demonstrated by applying a modified Gray-Thornton model to the experimental conditions. Empirical advection velocity and diffusion coefficient are proposed by fitting the model results to the experimental data and lead to shear rate dependent coefficients. This is different from Gray and Thornton who used constant values for dry granular flows. The last part of the paper describes 3D Discrete Element Method numerical simulations corresponding to the experimental conditions. The numerical simulations have been ran for about 60 s and qualitatively reproduce the grain size segregation at short time scale.

The paper is well written and provide very valuable experimental data as well as modeling guidelines on grain size segregation under bed load condition. The comparison with the advection-diffusion model of Gray and Thornton is also very relevant and allows to demonstrate the similarity of grain size segregation under bed-load

conditions with the dry granular case. It also highlights the differences such as the variable advection velocity and diffusion coefficient in space. The influence of the creep layer on the grain size segregation dynamic is quite new in the field of sediment transport and the authors data seems to show that this process could have an influence on river bed armoring at long timescales. I have more doubt about the significance of the DEM part which is rather qualitative. The authors could have done a lot more analysis on this part and it is quite frustrating for the reader at this point. I would recommend to either extend this part or remove it from the paper.

Reply: We thank the reviewer for her/his favorable and thorough review of our manuscript. We take the opportunity here to answer a concern raised by the reviewer about the extent of the DEM part of the study. Our main reason for performing dry DEM simulations has been to show that the fluid has little influence on segregation dynamics (by removing it, which the simulations do), and to back up our main conclusion that riverbed armoring follows the same phenomenology as of dry granular segregation. We believe the DEM part of the study has achieved this main goal in our mind. However, we now added two additional figures in supplementary materials with numerical simulations that further expand the DEM section and its significance for the study. In particular, Figure S10 in response to comment raised by reviewer #3 shows simulations with inclusion of a roughness layer at the base of the granular bed. The roughness layer is made of a layer of hexagonal close packed large grains fixed at their positions. Comparison of the horizontal velocity profile in simulations with and without roughness layer shows that the observation of creep to rapid flow transition holds in both systems. Changing the boundary roughness is difficult to do in experiments, because it requires significant modification to the channel and this could also negatively impact the optics that allow us to image grains. The magnitude of the horizontal velocity in the creep regime may decrease by an order in presence of boundary roughness, which indicates a higher dissipation rate at the boundary. In addition, Figure S11 shows the time-series of the Center of Mass of large grains in the DEM simulation of a bidisperse, sheared bed with different densities for the grains (with densities, ρ_{PMMA} , $\rho_{glassbead}$, $\rho_{PMMA} - \rho_{fluid}$, for PMMA, submerged PMMA and glassbead grains, respectively) again in response to the comment raised by reviewer #3. These simulations results show that submergence of grains, and their relative density compared to the fluid, have qualitatively no effect on the generality of our observations. Again, changing particle density in the experiment would be challenging because particles and fluid are refractive-index matched. The simulation allows us to do this easily.

As a conclusion of my review, I believe the paper should be published after the following questions and comments are addressed by the authors.

Questions and comments

1) Concerning the extended Gray-Thornton model, the authors claims that the advection and diffusion coefficients in the bed-load layer should depend on the shear rate but in their proposition the authors relate these coefficient to the vertical position. The parameter beta of the exponential function for the advection-diffusion coefficients is calibrated for each condition and seems to be quite sensitive. The dependency on the shear rate is rather indirect however I believe the model would gain generality if these coefficients where directly parameterized based on the local shear rate. I think some clarifications are needed here.

Reply: We would like to clarify here that in our extended Gray-Thornton model, depth-dependent advection-diffusion parameters ($S_{rn}(z)$ and $D_{rn}(z)$) follow the variation of shear velocity in the granular bed with depth. The parameter β is not a calibration, but it is the exponential decay constant of the fit to the variation of shear velocity with depth in each experimental run. We have shown earlier in the paper that shear velocity in the both bedload and creep regimes decays exponentially, but with different rate constants. Our model is developed based on the idea that advection and diffusion coefficients follow the same functional form and power of the velocity profile with depth in each experiment; therefore, the model inherently has a shear-rate dependency. In addition, since the derivative of an exponential function is the same exponential function, we can write $S_{rn}(z)$ and $D_{rn}(z)$ explicitly based on shear rate. In doing so, the results and formulations of the study hold entirely, with a small change in constants D_r and S_r , since the constant denominator in Eqs. 3 and 4, i.e. $[exp(\beta) - 1]$, must be incorporated into values of D_r and S_r . The ratio of S_{rn}/D_{rn} remains the same for all stresses.

2) In the same line, I would have liked to see the results of the Gray-Thornton model using constant advection-diffusion coefficient and also the sensitivity to the modeling of the creep regime using a constant diffusion coefficient. I wonder if the creep contribution is that important.

Reply: See Figure 1 below. We present comparison of the experimental concentration map (Fig. 1A) with: results of the advection-diffusion model as implemented in the manuscript (Fig. 1B); a model with constant advection-diffusion coefficient (same ratio $S_{rn}/D_{rn} \sim 318$ as in the paper) (Fig. 1C); and a model with a constant diffusion profile with depth, with same bedload ratio $S_{rn}/D_{rn} \sim 318$ as in the paper. Figure 1C provides compelling evidence that the constant advection-diffusion coefficient cannot produce the experimental profile. Moreover, we know from previous work by Fan et al. (2015) that the diffusion coefficient in the dense-rapid granular flow is indeed shear-rate dependent, and that it becomes shear-rate-independent in the creep regime. Therefore, a constant diffusion profile with depth — even with a depth-varying, shear-rate dependent advection profile as in Fig. 1D — is not successful in reproducing the experimental observation.

Figure 1: (A) 1D (x -averaged) concentration map of large grains over time for experiment under shear stress $\tau_s^* = 4.1\tau_{cs}^*$. (B) 1D (x -averaged) concentration map of large grains over time from the advection-diffusion model, with velocity profiles and initial condition corresponding to the shear stress $\tau_s^* = 4.1\tau_{cs}^*$ in panel (A), as is originally implemented in the manuscript. (C) Concentration map of large grains from the advection-diffusion model using constant advection and diffusion coefficients. (D) concentration map of large grains from the advection-diffusion model using the advection profile originally described in the paper, but with a constant diffusion coefficient along the depth.

3) Similar to what the authors have done for the advection time scale in the bed load regime, could the authors

Fan, Y., Umbanhowar, P. B., Ottino, J. M., & Lueptow, R. M. (2015). Shear-rate-independent diffusion in granular flows. *Physical review letters*, 115(8), 088001.

give an estimate of the time scale associated with the creep regime? This would help to estimate the influence of the creep regime to the segregation dynamic.

Reply: We have presented an estimate for the Peclet number, $Pe \sim 1.3$ in the bedload regime for experiment $\tau_s^* = 4.1\tau_{cs}^*$. The value of the Peclet number shows the ratio of advection rate to diffusion rate, and suggests that advective and diffusion are equally competing in bedload with a slight dominance of advection. As we move to the creep regime, the system is fully diffusion dominated and the Peclet number must tend to zero in the creep regime. However, in order to form an armor layer, all grains must go through the bedload layer in their journey to the surface and advective transport still plays a role on average — even when the bedload layer itself is depleted of large grains. So, the creep timescale depends somehow on the diffusivity of creeping grains, the concentration of coarse grains in the creep zone, and also on advection within the upper bedload layer. It seems difficult to provide a characteristic timescale for this.

4) The conclusions on the scaling analysis should be taken carefully as only one set of bidisperse mixture has been studied so far. In order to get more insight into the scaling laws for segregation more combinations of particle diameters, density and fluid viscosity should be investigated. I think the authors should add a sentence in the discussion to warn the readers.

Reply: We agree with the reviewer and have added words of caution in this part of the manuscript, regarding the scaling analysis and the comments raised by the reviewer .

5) It is not obvious to me how sensitive the results are to the creep regime in terms of segregation? Could the authors clarify the relative importance of bed-load and creep regime for segregation at long time scales.

Reply: At long timescales, meaning after the bedload layer is depleted of large grains, segregation and armoring take place only because persistent creep continues to deliver grains to the bedload layer. We would not expect to see longterm segregation without creep motion, so its importance is existential.

6) The choice of the restitution coefficient ($en=0.01$) should be justified based on the Stokes number. There exist some data in the literature that allows to do that (see Gondret et al. PoF 2002). This is only if the DEM part is kept in the future version of the manuscript.

Reply: We thank the reviewer for her/his remark. We actually chose the restitution coefficient based on the study by Gondret et al. PoF 2002, but had forgotten to cite the study appropriately. The value of restitution coefficient ($en=0.01$) used in the study corresponds to the range of Stokes number < 1 , and the reference for this is now cited appropriately in the manuscript.

7) In Fig 4D, it is hard to say that the DEM results agree with the experiments, at least for the creep influence on segregation as the model has not been run long enough. According to Fig 3, the creep influence occurs at time scale of about 10^4 s and the DEM run only lasts for 60s.

Reply: We agree with the reviewer that the simulations could have been run for longer time duration and provide more information. Currently, this is limit of our computational resources. However, we believe that the current time duration of the simulations is enough to cover the transition from bedload (dense rapid granular flow)-dominated segregation to the creep-dominated one. The transition in the experiments happens at $t \sim 30 - 300$ s for different stresses. In numerical DEM simulations, the transition occurs at $t \sim 10 - 50$ s. We believe that while the numerical simulations cover the transition, the slight difference in the transition time is mainly due to the distribution of large grains with depth in the two systems. In the experiments, the distribution is difficult to control based on the preparation protocol; as is clear from concentration maps, large grains are more concentrated in the creeping zone and are not uniform with depth. Whereas, in the numerical DEM simulation, the distribution of large grains is uniform across depth of the system.

Minor comment

The legend of some figures are too small. I think the authors should try to improve the plots, there is a lot of informations in each figure, maybe the can split some of them.

Reply: These are now corrected and the size of most figure legends are increased according to the formatting

Reviewer #3:

(Remarks to the Author):

The velocity profiles in Figures S4 through S6 show that there is slip between the plastic lower wall of the annulus and the lowermost spherical particles. The previous article in Nature Communications (Supp. Fig. 7) by most of the same authors shows the same phenomenon. The rate of shear between the wall and particles is much greater than between the particles in the creep zone. In fact S4 and S5 show essentially zero mean shear rate in the creep zone (i.e. plug flow). Perhaps the slip is not surprising given that the buoyant weight of the particles are more than an order of magnitude less than sediment in water. The rate of slip between the wall and particles increases systematically with shear stress. It is likely that the slip is not continuous in time and space (stick-slip) and leads to jostling of particles above. This could be the reason for the scatter in particle velocities. I was not able to obtain the supplemental videos for this paper, but the movies from the previous Nature Communications article show that settlement and shear of grains in the interior over the first several hours. The authors did not address how this time dependent behavior was dealt with for averaging. Were the first several hours removed from the analysis? The slip between the particles and bed is not addressed. I think it makes the conclusions of this paper (and the previous Nature Comm paper) dubious. The authors believe that the annulus setup to be superior to a flume with a downstream boundary. I have the opposite view. In fact, I would suggest installing ribs across the annulus wall to stop the slip. Another possibility is to use more dense spheres such as borosilicate. I recommend that the paper be rejected with encouragement of resubmission if the authors can fully address these issues.

Reply: We thank the reviewer for her/his cautious note. We first note that, as we pointed out above, changing the boundary roughness in the experiment is not trivial; this would require modifications to the channel that could influence the optics and our ability to image the bed. In addition, forcing a no-slip condition at the bottom boundary (such as with paddles, as the reviewer suggests) is not obviously advantageous; if creep is intentionally shut off at the bottom boundary in a channel of finite depth, this effect could influence much of the granular bed and suppress entirely an important dynamic. But, we agree with the reviewer that the issue of boundary roughness influence on these dynamics is potentially important and should be explored. Accordingly, we performed DEM simulations with inclusion of a fixed roughness layer at the base of the granular bed (new Figure S10 in the manuscript, and Figure 2 in this response letter). The roughness layer is made of a layer of hexagonal close packed large grains fixed at their positions to resemble the idea of “ribs” proposed by the reviewer. Comparison of the horizontal velocity profile in simulations with and without the roughness layer shows that the observation of creep to rapid flow transition holds in both systems. The magnitude of the horizontal velocity in the creep regime may decrease by an order in the presence of boundary roughness, which indicates a higher dissipation rate at the boundary.

The reviewer suggests that creep is somehow occurring because of the low submerged density of our grains compared to, say, quartz in water. As mentioned above, performing experiments with a different material (say borosilicate glass, as the reviewer suggests) is not feasible with our current setup, as our imaging technique relies on matching the refractive index of particles and fluid. But we can explore the influence of particle density easily with simulations. We have completed new simulations run and created a new supplementary Figure S11 in the manuscript (Figure 3 in the response letter), which shows the time-series of the Center of Mass of large grains in the DEM simulation of a bidisperse sheared bed with different densities for the grains (with densities, ρ_{PMMA} , $\rho_{glassbead}$, $\rho_{PMMA} - \rho_{fluid}$, for PMMA, submerged PMMA and glassbead grains, respectively). The simulations show that submergence of grains, and their relative density compared to the fluid, have qualitatively no effect on the generality of our observations.

The reviewer raises the issue of settling/compaction of the bed, and how this has been accounted for. There are two forms of initial settlement/compaction in the granular layer. The first form is a logarithmic settlement under gravity after the initial preparation — before the experiment itself begins — that happens mainly due to slippage and ageing of contacts. The second form is a settlement/compaction under shear. To avoid the influences of the first form, after the preparation phase of each experiment, the suspensions were left for ~ 30 minutes to allow time for this sedimentation, and relaxation and compaction of the granular bed, to reach completion. As for the second, in previous experiments with homogeneous particles we indeed waited for several hours of shear before taking measurements to ensure that data were collected in the steady state regime. This is impossible in the present bimodal experiments, however, since vertical segregation begins as soon as shear is applied. We try to minimize this effect by preparing the experiment under an initial shear phase that works to mix the grains. We however cannot neglect the first few hours of experiment/shearing.

Tapping or vibrating the granular bed is also not a solution since that would cause segregation by itself. We have made clear in the manuscript that this source of uncertainty influences rates of phenomenon. We however believe it is not a major concern, given that we have done our best to minimize the effect. Moreover, we point out the the continuum model does not include effects of shear-induced compaction, yet it is able to reproduce the experiments and the rate constants derived from the model are comparable to those determined from grain-scale measurements in the experiments. Finally, we point out a statement that was already in the supplementary which indicates that compaction may not be significant at all: “Slow granular compaction [refs] and slow dilation due to segregation [refs] approximately counterbalance such that the surface position remains almost constant as the armoring experiments progress.” Constant surface position means no net compaction

Finally, the reviewer suggests that creeping grains move as a “plug flow”, and that there is likely stick slip at the boundary which then jostles the grains above — and that this is the origin of creep. Our previous papers (Nature Communications 2015 and Physical Review E 2016) show clearly that creep is not a plug flow. Grain motion is localized and highly heterogeneous. In addition, the mean-square displacement measurements from grain trajectories support the idea that grain motion is localized and heterogeneous; grains show caging at smaller timescales and a transition to diffusive motion. A plug flow would produce mean-square displacements that scale ballistically — this is not what is observed. Moreover, there are no observations that support a large-scale stick slip motion at the bottom boundary. The simulations with a rough boundary, which were added in response to the reviewer, of course produce a slow-down of creep — especially near the bottom rough boundary. However, creep still occurs and is distinct from the dense-granular (bedload) flow above, and is qualitatively similar to that which occurs with a smooth boundary. Moreover, we point out that sub-critical creep in granular systems is very commonly observed and does not appear to require a special system (see refs within our paper).

- Houssais, M., Ortiz, C. P., Durian, D. J., & Jerolmack, D. J. (2015). Onset of sediment transport is a continuous transition driven by fluid shear and granular creep. *Nature communications*, 6.

- Houssais, M., Ortiz, C. P., Durian, D. J., & Jerolmack, D. J. (2016). Rheology of sediment transported by a laminar flow. *Physical Review E*, 94(6), 062609.

Figure 2: (A) A snapshot from the armored/segregated state of a simulation with a roughness layer at its base. The simulation is run with $u_{top} = 0.08 \text{ m/s}$. (B) Velocity profiles for simulations with ("w/ br" in legend) and without ("w/o br" in legend) the roughness layer, at two surface layer velocities, u_{top} , of 0.05 and 0.08 m/s.

Figure 3: The time-series of the Center of Mass of large grains in DEM simulation of bidisperse sheared bed with different densities for the grains (with densities, ρ_{PMMA} , $\rho_{glassbead}$, $\rho_{PMMA} - \rho_{fluid}$, for PMMA, submerged PMMA and glassbead grains, respectively).

Reviewers' comments:

Reviewer #1 (Remarks to the Author):

By performing some additional simulation exercises the authors have shown that there are reasonable grounds to suppose that their results are robust to variations in certain properties of the experimental materials and procedures. This has strengthened the force of the conclusions. I recommend publication of the present paper.

Reviewer #2 (Remarks to the Author):

Referee report on « River-bed armoring as a granular segregation phenomenon » for publication in Nature Communications by B. Ferdowsi, C.P. Ortiz, M Houssais and D.J. Jerolmack

Based on my reading of the revised manuscript and the authors rebuttal, I am still not convinced by part of the work, the DEM part of the work is really too weak. Also, I find regrettable that so little of the authors answers have been added to the revised manuscript. Most of the modifications have been done in the supplementary information however some of their answers could increase the impact of the work (sensitivity to the advection-diffusion coefficients - question 2) . At this stage I would not recommend publication but I would be happy to read a new version of the manuscript taking into account my major concern about the DEM simulations.

Questions/answers from my previous review

1) Concerning the extended Gray-Thornton model, the authors claims that the advection and diffusion coefficient in the bed-load layer should depend on the shear rate but in their proposition the authors relate these coefficient to the vertical position. The parameter beta of the exponential function for the advection-diffusion coefficients is calibrated for each condition and seems to be quite sensitive. The dependency on the shear rate is rather indirect however I believe the model would gain generality if these coefficients were directly parameterized based on the local shear rate. I think some clarifications are needed here.

Reply: We would like to clarify here that in our extended Gray-Thornton model, depth-dependent advection/diffusion parameters ($S_{rn}(z)$ and $D_{rn}(z)$) follow the variation of shear velocity in the granular bed with depth. The parameter is not a calibration, but it is the exponential decay constant of the fit to the variation of shear velocity with depth in each experimental run. We have shown earlier in the paper that shear velocity in the both bedload and creep regimes decays exponentially, but with different rate constants. Our model is developed based on the idea that advection and diffusion coefficients follow the same functional form and power of the velocity profile with depth in each experiment; therefore, the model inherently has a shear-rate dependency. In addition, since the derivative of an exponential function is the same exponential function, we can write $S_{rn}(z)$ and $D_{rn}(z)$ explicitly based on shear rate. In doing so, the results and formulations of the study hold entirely, with a small change in constants D_r and S_r , since the constant denominator in Eqs. 3 and 4, must be incorporated into values of D_r and S_r . The ratio of $S_{rn}=D_{rn}$ remains the same for all stresses.

R#2: Thank you for the clarification but you never give the actual values of beta. You should add in Fig S4-S8 the fitted velocity profiles and explicitly give the fitted beta values for both layers.

—

2) In the same line, I would have liked to see the results of the Gray-Thornton model using constant advection-diffusion coefficient and also the sensitivity to the modeling of the creep regime using a constant diffusion coefficient. I wonder if the creep contribution is that important.

Reply: See Figure 1 below. We present comparison of the experimental concentration map (Fig. 1A) with: results of the advection-diffusion model as implemented in the manuscript (Fig. 1B); a model with constant advection-diffusion coefficient (same ratio $S_{rn}=D_{rn} \square 318$ as in the paper) (Fig. 1C); and a model with a constant diffusion profile with depth, with same bedload ratio $S_{rn}=D_{rn} \square 318$ as in the paper. Figure 1C provides compelling evidence that the constant advection-diffusion coefficient cannot produce the experimental profile. Moreover, we know from previous work by Fan et al. (2015) that the diffusion coefficient in the dense-rapid granular flow is indeed shear-rate dependent, and that it becomes shear-rate-independent in the creep regime. Therefore, a constant diffusion profile with depth — even with a depth-varying, shear-rate dependent advection profile as in Fig. 1D — is not successful in reproducing the experimental observation.

R#2: I would like to thank the authors for the extensive reply. I agree that the case shown in Fig 1C (constant advection and diffusion segregation fluxes) demonstrate that the advection flux should, as expected, depend on the local shear rate. However, the differences between Fig 1B and Fig 1D is not that important. I believe that a constant diffusion works as well. Is there a more quantitative way of demonstrating the authors point of view that the diffusion coefficient should also depend on the shear rate? From Fig1 this conclusion is not so clear to me. Also, I would recommend to include the new figure 1 in place of figure 3 of the revised manuscript.

—

3) - 6)

R#2: I am satisfied by the authors response

—

7) In Fig 4D, it is hard to say that the DEM results agree with the experiments, at least for the creep influence on segregation as the model has not been run long enough. According to Fig 3, the creep influence occurs at time scale of about 10^4 s and the DEM run only last for 60s.

Reply: We agree with the reviewer that the simulations could have been run for longer time duration and provide more information. Currently, this is limit of our computational resources. However, we believe that the current time duration of the simulations is enough to cover the transition from bedload (dense rapid granular flow)-dominated segregation to the creep-dominated one. The transition in the experiments happens at $t \square 30 - 300$ s for different stresses. In numerical DEM simulations, the transition occurs at $t \square 10 - 50$ s. We believe that while the numerical simulations cover the transition, the slight difference in the transition time is mainly due to the distribution of large grains with depth in the two systems. In the experiments, the distribution is difficult to control based on the preparation protocol; as is clear from concentration maps, large grains are more concentrated in the creeping zone and are not uniform with depth. Whereas, in the numerical DEM simulation, the distribution of large grains is uniform

across depth of the system.

R#2: The authors' answer is not satisfying. I insist that comparing the DEM simulation results with the experiments over 60s is not enough when the segregation process is shown to last over 10^2 to 10^3 s experimentally. The authors have to perform at least one run over a longer duration. Moreover, the authors only show the time evolution of the center of mass, they should at least compare the velocity profiles obtained from the DEM simulations as the shear rate is controlling the segregation process in the bed load layer. The new runs added to the supplementary information only last 5s! I maintain my major comment from the first round, if the authors can not improve the DEM results due to computational resources they should remove it from the manuscript.

Minor comment

Please use always the same units in your graph, meter or millimeter and do not change between figures.

P.6 - l.114: correct 9ore to more

P.15 - l.317-318: The conclusion: « Results suggest minimal influence of the fluid beyond determining the surface grain velocity. » should be removed as the DEM simulations are not reliable.

P.16-l.349-350: I don't think the authors can say that polydisperse situation is amenable to DEM simulations as the authors are not able to run a case for more than 60s in the bidisperse situation. The polydisperse case would be even more computationally demanding as the number of particles in the simulation would grow by a huge factor.

Response to reviewers

We present in this document point-by-point response to reviewers' comments and suggestions with describing the changes made in the manuscript to implement those suggestions. The changes in the main paper are highlighted in reply to each point. Several significant changes and additions have been made to the main manuscript in order to address questions raised by the 2nd reviewer. We now ran all our DEM simulations for 1160 seconds and prepared new Figure 5D for the main manuscript and Figure S11 for supplementary materials with these significantly longer simulations. We hope these revisions would satisfy concerns of the reviewer regarding DEM simulations presented in the paper and some of the conclusions of the manuscript. Regarding our extended version of advection-diffusion model, we now have calculated the temporal evolution of armor layer thickness for different versions of the model, previously only shown in the response letter, and added it as a new panel to Figure 3 in the main manuscript. We also added our responses to the 2nd reviewer Questions (1) and (2) to the main manuscript. The response to question (2) accompanies the description and explanation on the new Fig. 3 in the main manuscript and the response to question (1) is added as the 3rd paragraph in the *Discussion* section of the manuscript. We have further corrected the consistency problems with units of some of the figures, removed some of the questionable (according to the 2nd reviewer) statements from our conclusions and discussions, and corrected several typos.

Reviewer Comments:

Reviewer #1:

Reviewer's Reply 2nd round: By performing some additional simulation exercises the authors have shown that there are reasonable grounds to suppose that their results are robust to variations in certain properties of the experimental materials and procedures. This has strengthened the force of the conclusions. I recommend publication of the present paper.

Authors' Reply 2nd round: We thank the reviewer for his positive remarks and encouraging and thorough review.

Reviewer #2:

Referee report on «River-bed armoring as a granular segregation phenomenon» for publication in *Nature Communications* by B. Ferdowsi, C.P. Ortiz, M Houssais and D.J. Jerolmack

Reviewer's General comment 2nd round: Based on my reading of the revised manuscript and the authors rebuttal, I am still not convinced by part of the work, the DEM part of the work is really too weak. Also, I find regrettable that so little of the authors answers have been added to the revised manuscript. Most of the modifications have been done in the supplementary information however some of their answers could increase the impact of the work (sensitivity to the advection-diffusion coefficients - question 2) . At this stage I would not recommend publication but I would be happy to read a new version of the manuscript taking into account my major concern about the DEM simulations. .

Authors' Reply 2nd round: We thank the reviewer for his/her remarks. Based on general concerns of the reviewer, we now have added our response to his/her previous question 2 on our extended version of advection-diffusion model as a new Fig. 3 in the main manuscript. We have further added our response to his/her previous question on shear-rate dependency of our extended version of advection-diffusion model to the Discussion part of the manuscript (Paragraph 3). Additionally, we have run all our DEM simulation now for 1160 seconds and updated Figure 5D in the main manuscript and Fig. S11 in the supplementary materials with new significantly longer simulations that go well beyond the transitional point from bedload-dominated to creep-dominated segregation. We hope these revisions would satisfy concerns of the reviewer.

Questions and comments

1) Concerning the extended Gray-Thornton model, the authors claims that the advection and diffusion coefficients in the bed-load layer should depend on the shear rate but in their proposition the authors relate these coefficient to the vertical position. The parameter beta of the exponential function for the advection-diffusion coefficients is calibrated for each condition and seems to be quite sensitive. The dependency on the shear rate is rather indirect however I believe the model would gain generality if these coefficients were directly parameter-

Figure 1: New Figure 3 of the main manuscript: (A) 1D (x -averaged) concentration map of large grains over time for experiment under shear stress $\tau_s^* = 4.1\tau_{cs}^*$. (B) 1D (x -averaged) concentration map of large grains over time from the advection-diffusion model, with velocity profiles and initial condition corresponding to the shear stress $\tau_s^* = 4.1\tau_{cs}^*$ in panel (A), as is originally implemented in the manuscript. (C) Concentration map of large grains from the advection-diffusion model using constant advection and diffusion coefficients. (D) concentration map of large grains from the advection-diffusion model using the advection profile originally described in the paper, but with a constant diffusion coefficient along the depth. (E) Temporal evolution of the thickness of the armored layer at shear stress $\tau_s^* = 4.1\tau_{cs}^*$ calculated from experimental observation corresponding to (A) and different implementations of the advection-diffusion model that their concentration maps are shown in (B)-(D).

ized based on the local shear rate. I think some clarifications are needed here.

Authors' Reply 1st round: We would like to clarify here that in our extended Gray-Thornton model, depth-dependent advection-diffusion parameters ($S_{rn}(z)$ and $D_{rn}(z)$) follow the variation of particle velocity in with depth. The parameter β is not a calibration, but it is the exponential decay constant of the fit to the velocity profile for each experimental run. We have shown earlier in the paper that shear velocity in both the bed load and creep regimes decays exponentially, but with different rate constants. Our model is developed based on the idea that advection and diffusion coefficients follow the same functional form and power of the velocity profile with depth in each experiment; therefore, the model inherently has a shear-rate dependency. In addition, since the derivative of an exponential function is the same exponential function, we can write $S_{rn}(z)$ and $D_{rn}(z)$ explicitly based on shear rate. In doing so, the results and formulations of the study hold entirely, with a small change in constants D_r and S_r , since the constant denominator in Eqs. 3 and 4, i.e. $[\exp(\beta) - 1]$, must be incorporated into values of D_r and S_r . The ratio of S_{rn}/D_{rn} remains the same for all stresses.

Reviewer's Reply 2nd round: Thank you for the clarification but you never give the actual values of beta. You should add in Fig S4-S8 the fitted velocity profiles and explicitly give the fitted beta values for both layers.

Authors' Reply 2nd round: We thank the reviewer for his/her remark. We now have added the fitted velocity profiles for the bed load layers (that have advective-dominated segregation mechanism) to the supplementary figures S4-S8. The value of β and the relevant velocity profile equations are also shown as the legend of each figure in Figs. S4-S8. The values of β at each stress are also reported in Table S1, column #2. We assume a constant diffusion coefficient in the creep layer below bedload and do not fit the velocity profile and do not measure β values for the creep layers. We also have considered again the general remark of the reviewer and added our previous response about shear versus velocity dependence of our extended advection-diffusion model to the main manuscript. Please see the 3rd paragraph in the *Discussion* section.

2) In the same line, I would have liked to see the results of the Gray-Thornton model using constant advection-diffusion coefficient and also the sensitivity to the modeling of the creep regime using a constant diffusion coefficient. I wonder if the creep contribution is that important.

Authors' Reply 1st round: See Figure 1 below. We present comparison of the experimental concentration map (Fig. 1A) with: results of the advection-diffusion model as implemented in the manuscript (Fig. 1B); a model with constant advection-diffusion coefficient (same ratio $S_{rn}/D_{rn} \sim 318$ as in the paper) (Fig. 1C); and a model with a constant diffusion profile with depth, with same bedload ratio $S_{rn}/D_{rn} \sim 318$ as in the paper. Figure 1C provides compelling evidence that the constant advection-diffusion coefficient cannot produce the experimental profile. Moreover, we know from previous work by Fan et al. (2015) that the diffusion coefficient in the dense-rapid granular flow is indeed shear-rate dependent, and that it becomes shear-rate-independent in the creep regime. Therefore, a constant diffusion profile with depth — even with a depth-varying, shear-rate dependent advection profile as in Fig. 1D — is not successful in reproducing the experimental observation.

Reviewer's Reply 2nd round: I would like to thank the authors for the extensive reply. I agree that the case shown in Fig 1C (constant advection and diffusion segregation fluxes) demonstrate that the advection flux should, as expected, depend on the local shear rate. However, the differences between Fig 1B and Fig 1D is not that important. I believe that a constant diffusion works as well. Is there a more quantitative way of demonstrating the authors point of view that the diffusion coefficient should also depend on the shear rate? From Fig1 this conclusion is not so clear to me. Also, I would recommend to include the new figure 1 in place of figure 3 of the revised manuscript.

Authors' Reply 2nd round: We thank the reviewer for his/her remarks. We now have added our response to his/her previous question 2 to the main manuscript and used Fig. 1 of the response letter as a new Fig. 3 in the main manuscript. Further, we added a new panel (E) to Fig. 3 of the main manuscript that compares the temporal evolution of the armor layer thickness and segregation progress with three versions of the advection-diffusion model (1-constant advection-diffusion coefficients, 2-depth-varying advection but a constant diffusion coefficient and 3-rate-dependent advection-diffusion coefficients). Panel (E) of Fig. 3 provides a more quantitative comparison of the thickness of the armored layer through time and indicates

that the model with rate-dependent advection-diffusion coefficients has superior predictive power, and also matches the data well for the entire range of Shields stresses (Fig. 4A of the main manuscript). Importantly, the model correctly captures the initial fast and subsequent slow stages of segregation.

Remarks 3 - 6

Reviewer's Reply 2nd round: I am satisfied by the authors response.

Authors' Reply 2nd round: Thank you.

7) In Fig 5D, it is hard to say that the DEM results agree with the experiments, at least for the creep influence on segregation as the model has not been run long enough. According to Fig 3, the creep influence occurs at time scale of about 10^4 s and the DEM run only lasts for 60s.

Authors' Reply 1st round: We agree with the reviewer that the simulations could have been run for longer time duration and provide more information. Currently, this is limit of our computational resources. However, we believe that the current time duration of the simulations is enough to cover the transition from bedload (dense rapid granular flow)-dominated segregation to the creep-dominated one. The transition in the experiments happens at $t \sim 30 - 300$ s for different stresses. In numerical DEM simulations, the transition occurs at $t \sim 10 - 50$ s. We believe that while the numerical simulations cover the transition, the slight difference in the transition time is mainly due to the distribution of large grains with depth in the two systems. In the experiments, the distribution is difficult to control based on the preparation protocol; as is clear from concentration maps, large grains are more concentrated in the creeping zone and are not uniform with depth. Whereas, in the numerical DEM simulation, the distribution of large grains is uniform across depth of the system.

Reviewer's Reply 2nd round: The authors' answer is not satisfying. I insist that comparing the DEM simulation results with the experiments over 60s is not enough when the segregation process is shown to last over 10^2 to 10^3 s experimentally. The authors have to perform at least one run over a longer duration. Moreover, the authors only show the time evolution of the center of mass, they should at least compare the velocity profiles obtained from the DEM simulations as the shear rate is controlling the segregation process in the bed load layer. The new runs added to the supplementary information only last 5s! I maintain my major comment from the first round, if the authors can not improve the DEM results due to computational resources they should remove it from the manuscript.

Authors' Reply 2nd round: We thank the reviewer for his/her comments and suggestions on the DEM simulations part of the work. We now have run DEM simulations for 1160 seconds at two surface velocities. The results from these simulations have now replaced the previous results in Fig. 5D. The patterns and behavior is very much similar to the experimental observations, showing two phases of armor formation: a first fast and advection-dominated segregation/armoring phase, followed by a creep-driven slow armor-ing/segregation phase. Furthermore, we also have run the numerical DEM simulations for investigating the influences of grains density for 1160 seconds and replaced the Center of Mass measurements that were presented in the previous version of the paper (Fig. S11) with velocity profiles in systems with different grain densities for the current version. The new results presented in Fig. S11 indicate little change in the shear velocity and shear rate variations and consequently little influence on the segregation dynamics as a result of changing the grain densities uniformly for all grain sizes.

Minor comment

Reviewer's Reply 2nd round:

Point (1) Please use always the same units in your graph, meter or millimeter and do not change between figures.

Point (2) P.6 - l.114: correct 9ore to more

Point (3) P.15 - l.317-318: The conclusion: "Results suggest minimal influence of the fluid beyond determining the surface grain velocity." should be removed as the DEM simulations are not reliable.

Point (4) P.16-l.349-350: I don't think the authors can say that polydisperse situation is amenable to DEM simulations as the authors are not able to run a case for more than 60s in the bidisperse situation. The polydisperse case would be even more computationally demanding as the number of particles in the simula-

Fan, Y., Umbanhowar, P. B., Ottino, J. M., & Lueptow, R. M. (2015). Shear-rate-independent diffusion in granular flows. *Physical review letters*, 115(8), 088001.

New Fig 5D

Figure 2: New Fig. 5D for DEM simulations ran for 1160 seconds. The new figure shows the evolution of armor thickness for the DEM model and experiments at two equivalent driving stresses.

Figure 3: New Fig. S11 for DEM simulations ran for 1160 seconds with different grain densities. We replaced Center of Mass measurements with velocity profiles according to the suggestion of the reviewer.

tion would grow by a huge factor.

Authors' Reply 2nd round: Point (1) and Point (2) are now corrected in the manuscript. All units are now consistent. Regarding point (3) and with new longer DEM simulations we adhere to our previous conclusion. Regarding point (4), we have removed those sentences from the manuscript per suggestion of the reviewer.

REVIEWERS' COMMENTS:

Reviewer #2 (Remarks to the Author):

I have now read the revised manuscript and authors' answers and I am satisfied with their answers and modifications to the manuscript. In my view the paper can be accepted for publication.